# EPDR1 promotes PD-L1 expression and tumor immune evasion by inhibiting TRIM21-dependent ubiquitylation of IkappaB kinase-β

Xiaoyu Qian [ID][1,3], Jin Cai [ID][1,3], Yi Zhang[1], Shengqi Shen[2], Mingjie Wang[1], Shengzhi Liu [ID][1], Xiang Meng[1], Junjiao Zhang[1], Zijian Ye[1], Shiqiao Qiu[1], Xiuying Zhong [ID][2]✉ & Ping Gao [ID][1,2]✉

## Abstract

**While immune checkpoint blockade (ICB) has shown promise for clinical cancer therapy, its efficacy has only been observed in a limited subset of patients and the underlying mechanisms regulating innate and acquired resistance to ICB of tumor cells remain poorly understood. Here, we identified ependymin-related protein 1 (EPDR1) as an important tumor-intrinsic regulator of PD-L1 expression and tumor immune evasion. Aberrant expression of EPDR1 in hepatocellular carcinoma is associated with immunosuppression. Mechanistically, EPDR1 binds to E3 ligase TRIM21 and disrupts its interaction with IkappaB kinase-b, suppressing its ubiquitylation and autophagosomal degradation and enhancing NF-κB-mediated transcriptional activation of PD-L1. Further, we validated through a mouse liver cancer model that EPDR1 mediates exhaustion of CD8+ T cells and promotes tumor progression. In addition, we observed a positive correlation between EPDR1 and PD-L1 expression in both human and mouse liver cancer samples. Collectively, our study reveals a previously unappreciated role of EPDR1 in orchestrating tumor immune evasion and cancer progression.**

**Keywords** Tumor Immune Evasion; EPDR1; TRIM21; Hepatocellular Carcinoma
**Subject Categories** Cancer; Immunology

## Introduction

Hepatocellular carcinoma (HCC) is one of the most prevalent cancer types globally and is currently the third leading cause of cancer-related death (Cronin et al, 2022; Sung et al, 2021). The treatment of HCC remains a significant healthcare challenge worldwide. Immune checkpoint blockade (ICB) has the potential to disrupt the immune escape of cancer cells by removing inhibitory signals of T-cell activation, thus enhancing patient survival in various cancer types, making it a promising clinical antitumor therapy. However, it has only been shown to be effective in a subset of patients (Llovet et al, 2021; Morad et al, 2021; Topalian et al, 2023). Identifying patients who may benefit from ICB is crucial for the precise application of these drugs. However, there is currently no effective method to achieve this.

The expression levels of checkpoint molecules, including PD-L1, are correlated with the effectiveness of ICB treatment (Yi et al, 2018; Yu et al, 2016). While researchers recognize that PD-L1 could be regulated in multiple ways, the identification of key molecules and the underlying mechanisms that drive the expression of PD-L1 and promote immune evasion in tumor-intrinsic and tumor microenvironments continue to elude researchers, impeding the broader application and maximum clinical benefits of cancer immunotherapies (Ghorani et al, 2023). Through transcriptome analysis of different subclasses of liver cancer patients, we identified novel oncogenes as potential markers for identifying patients who may benefit from ICB therapy. Among these is ependymin-related protein 1 (EPDR1). EPDR1, a relatively uncharacterized protein, has been found in the secretome of adipocytes and contributes to thermogenic determination during adipogenesis (Deshmukh et al, 2019). EPDR1 deficiency in β-cells impairs the glucose-stimulated increase in mitochondrial respiration and ATP/ADP-ratio, suggesting the important role of EPDR1 in regulating β-cell metabolism and function (Cataldo et al, 2022). Recent studies also suggest that EPDR1 exhibits aberrant expression in various cancers. Despite being related to some physiological and pathological processes, the molecular functions of EPDR1 in tumors are poorly understood (Chen and Zhang, 2020; Liang et al, 2020; Nimmrich et al, 2001; Riffo-Campos et al, 2016; Wei et al, 2019).

In this study, we set out to search for potential targets for liver cancer ICB treatment and find an unappreciated role for EPDR1 in regulating cancer progression and immune evasion. Using HCC mouse models, we found that tumor-intrinsic EPDR1 promotes CD8+ T-cell exhaustion by elevating the transcriptional level of PD-L1. Further mechanistic studies revealed that EPDR1 binds to the E3 ligase TRIM21 and inhibits the interaction between TRIM21 and IKBKB, which stabilizes the IKBKB protein and induces NF-κB-mediated PD-L1 transactivation. These findings suggest important therapeutic implications for EPDR1 in regulating antitumor immunity and cancer progression.

[1]School of Medicine, South China University of Technology, Guangzhou, China. [2]Medical Research Institute, Guangdong Provincial People's Hospital, Guangdong Academy of Medical Sciences, Southern Medical University, Guangzhou, China. [3]These authors contributed equally: Xiaoyu Qian, Jin Cai. ✉E-mail: zxywawj@ustc.edu.cn; pgao2@ustc.edu.cn

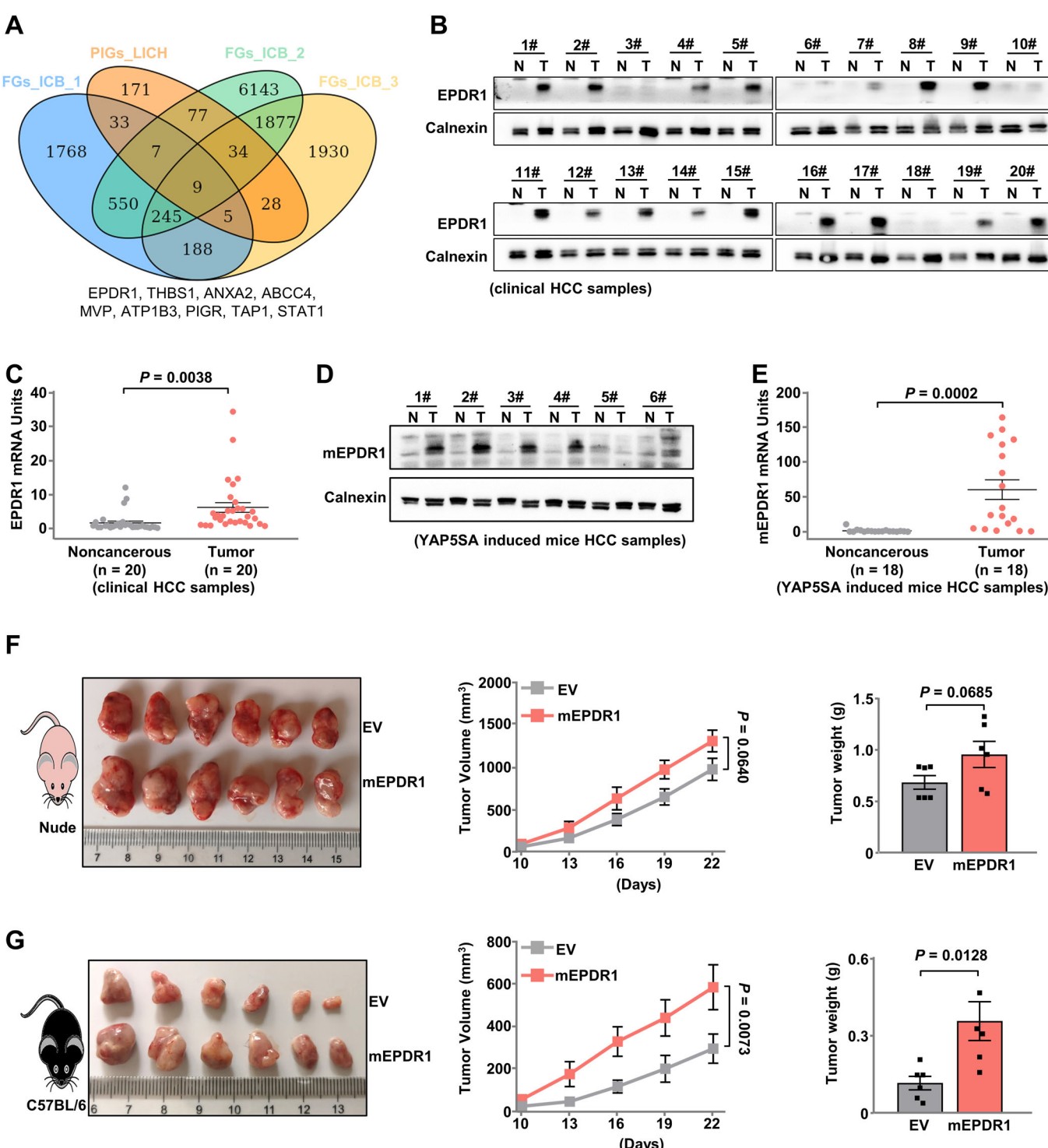

## Results

### Aberrant expression of EPDR1 in HCC is associated with immunosuppression

To elucidate the underlying association between gene patterns and the curative effects of ICB, we conducted a comprehensive review of multiple studies that categorized liver cancer patients into distinct subtypes, considering factors such as immune infiltration (Murai et al, 2023; Yoon et al, 2021) and intratumor heterogeneity (Yang et al, 2022); we performed an intersection analysis of characteristic genes associated with potential benefits of immune therapy and oncogenes that are progressively upregulated during liver cancer development (Zhu et al, 2023), and identified nine potential candidate molecules (Fig. 1A), some of which are well-known for their involvement in the immune system or have been identified as genes related to antitumor

**Figure 1.  Aberrant expression of EPDR1 in HCC is associated with immunosuppression.**

(A) Venn diagram illustrating the intersection between the four cohorts shows feature genes of patients who may benefit from ICB (FGs_ICB) and a combined analysis of progressively increased genes during liver cancer development (PIGs_LICH). (B) Immunoblotting analysis of EPDR1 protein levels in 20 pairs of clinically matched adjacent noncancerous liver tissues (Noncancerous) and human liver cancer tissues (Tumor). Calnexin served as a loading control. (C) EPDR1 mRNA levels were measured in 20 pairs of clinically matched tumor-adjacent noncancerous liver tissues (Noncancerous) and human HCC tissues (Tumor) ($n = 20$ patients with HCC). Data are presented as the mean ± SEM, and statistical analyses were performed by two-tailed unpaired Student's $t$-test. (D) Immunoblotting analysis of mEPDR1 protein levels in 6 pairs of matched tumor-adjacent noncancerous liver tissues (Normal) and liver cancer tissues (Tumor) from the YAP5SA-induced mouse model. Calnexin was used as a loading control. (E) mEPDR1 mRNA levels were measured in 18 pairs of matched tumor-adjacent noncancerous liver tissues (Normal) and liver cancer tissues (Tumor) from the YAP5SA-induced mouse model ($n = 18$ mice with HCC). Data were presented as the mean ± SEM, and statistical analyses were performed by two-tailed unpaired Student's $t$-test. (F) Equal numbers of Hepa 1-6 cells expressing Flag-EV or Flag-mEPDR1 were subcutaneously injected into immunodeficient nude mice ($n = 6$ mice in each group). Tumor size was measured starting at 10 days after inoculation. Photographs show xenografts (left), growth curves (middle, two-way ANOVA), and tumor weights (right, two-tailed unpaired Student's $t$-test) determined at the end of the experiment (day 25). Data were presented as the mean ± SEM. (G) Equal numbers of Hepa 1-6 cells expressing Flag-EV or Flag-mEPDR1 were subcutaneously injected into immunocompetent C57BL/6 mice ($n = 6$ mice in each group). Tumor size was measured starting at 10 days after inoculation. Photographs show xenografts (left), growth curves (middle, two-way ANOVA), and tumor weights (right, two-tailed unpaired Student's $t$-test) determined at the end of the experiment (day 25). Data were presented as the mean ± SEM. Source data are available online for this figure.

immunity in other screening studies, such as STAT1, PIGR, and TAP1 (Dubrot et al, 2022; Lawson et al, 2020; Pan et al, 2018; Upadhyay et al, 2021). This implies the reliability of our study. EPDR1 was ultimately chosen as the subject of our research due to the limited studies on cancer immunity. Analysis of the TCGA liver cancer dataset revealed a significant increase in EPDR1 expression in malignant tissue compared to healthy tissue (Fig. EV1A). Moreover, the expression of EPDR1 was significantly associated with reduced five-year survival rates in liver cancer patients (Fig. EV1B). We observed a significant increase in the protein level (Fig. 1B) and mRNA level (Fig. 1C) of EPDR1 in human HCC tissues compared to those in adjacent noncancerous tissues. This finding is consistent with the results obtained from mouse HCC samples induced by YAP5SA (Fig. 1D,E). The findings indicate that EPDR1 plays a critical role as an oncogenic factor in the pathogenesis of HCC.

To evaluate the effect of EPDR1 on the in vivo growth of liver cancer, we utilized an HCC mice model established by subcutaneous injection of Hepa 1–6 murine liver cancer cells expressing Flag-tag empty vector (Flag-EV) or murine EPDR1 (Flag-mEPDR1). In the nude mice, high expression of mEPDR1 conferred an advantage in terms of tumor development, although there was no significant difference (Fig. 1F); however, compared to control cells, mEPDR1-overexpressing cells exhibited significantly enhanced proliferation in immunocompetent mice (Fig. 1G). These findings suggest that EPDR1 may play a role in promoting HCC progression by regulating antitumor immunity.

## Tumor-intrinsic EPDR1 facilitates immune evasion by increasing antitumor CD8[+] T cells exhaustion

To investigate the impact of EPDR1 on tumor immunity, we employed a YAP5SA-induced mouse liver cancer model. Interestingly, the overexpression of mEPDR1 significantly aggravated the disease in terms of pathological features and hepatic tumor sizes compared to that in the control group (Figs. 2A,B and EV2A). Based on the expression of characteristic markers, we identified specific cell types within the clusters (Figs. 2C and EV2B). Our findings revealed a significant increase in the proportion of CD8[+]PD-1[+] cells in the mEPDR1 overexpression group (Fig. 2D), suggesting enhanced exhaustion of CD8[+] T cells and potential impairment of the cytotoxic effects (Beltra et al, 2020). Further flow cytometric results revealed a significantly increased proportion of exhausted CD8[+] T cells (i.e., PD-1[+], TIM-3[+], or PD-1[+]TIM-3[+]) and an

evident decrease in CD8[+] T-cell activity (indicated by the proportion of IFNγ[+] or Granzyme B[+] (GzmB[+]) cells) in HCC-bearing mice overexpressing mEPDR1 compared to tumor-bearing control mice (Figs. 2E and EV2C). Moreover, we conducted an in vitro coculture experiment to explore the direct impact of EPDR1 upregulation in tumor cells on the phenotype and function of CD8[+] T cells (Fig. 2F). Flow cytometric analysis revealed that CD8[+] T cells cocultured with EPDR1-overexpressing HepG2 cells exhibited increased PD-1 and TIM-3 expression, while IFNγ and GzmB expression decreased (Fig. 2G). Additionally, similar assays were performed using murine CD8[+] T cells and the hepatocarcinoma cell line Hepa 1–6, yielding results consistent with those coculture experiments involving human cells (Fig. EV2D,E). These findings suggest that the oncogenic effect of EPDR1 may influence the CD8[+] T-cell population within the tumor microenvironment.

To validate the role of CD8[+] T cells in the pro-carcinogenic effects of EPDR1, we used a CD8 monoclonal neutralizing antibody (α-CD8) to treat immune-competent mice that were subcutaneously inoculated with murine EPDR1- or empty vector-expressing Hepa 1–6 cells. Analysis of peripheral blood samples revealed significant depletion of CD8[+] T cells in the α-CD8 antibody-treated groups (Fig. EV2F). Subsequent examination showed no significant difference in tumor volume between the groups after neutralization of CD8[+] T cells (Fig. 2H). Evaluation of tumor morphology, final weight in the liver indicated that the growth advantage conferred by mEPDR1 overexpression in transplanted tumors was nullified upon CD8[+] T-cell neutralization (Fig. 2I,J). By employing a xenograft model generated by subcutaneous inoculation with murine liver cancer cell Hepa 1–6 expressing shEPDR1, we unveiled that suppression of EPDR1 led to a decrease in xenograft proliferation (Fig. EV2G–I). Additional flow cytometry analysis unveiled that the suppression of EPDR1 in cancer cells led to a reduction in exhaustion and an increase in the activation of tumor-infiltrating T cells in mice (Fig. EV2J,K). These findings suggest that CD8[+] T cells play a crucial role in mediating the pro-oncogenic phenotype induced by EPDR1. This result is consistent with our findings and further supports the notion that EPDR1 promotes tumor growth by exacerbating CD8[+] T-cell exhaustion.

## EPDR1 enhances the NF-κB pathway and elevates PD-L1 expression in cancer cells by interacting with TRIM21

To investigate the role of EPDR1 in regulating CD8[+] T-cell-dependent tumor immune evasion, we performed qRT-PCR analyses on a series

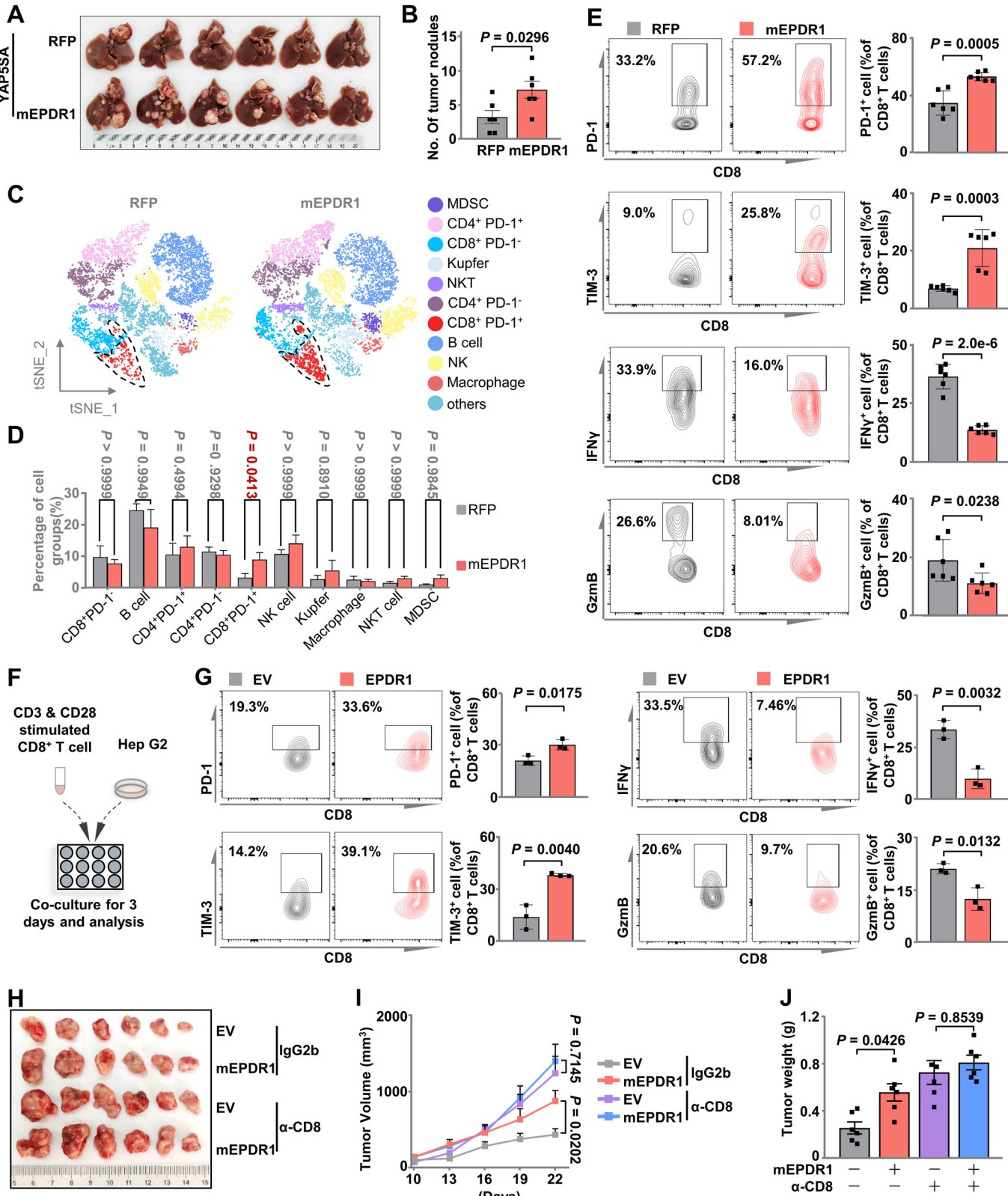

**Figure 2. Tumor-intrinsic EPDR1 facilitates immune evasion by increasing antitumor CD8+ T cells exhaustion.**

(A, B) Plasmids expressing human YAP5SA (2 mg/kg) plus RFP control (2 mg/kg) or mEPDR1 (2 mg/kg), together with plasmids expressing PB transposase (0.8 mg/kg), were delivered into mice by hydrodynamic injection (n = 6 male mice per group). Liver tumors were analyzed 110 days after injection. Photographs show livers (A) and tumor numbers (B) were determined. Data were presented as the mean ± SEM. (C) Dimensionality reduction and visualization based on T-distributed stochastic neighbor embedding (t-SNE) analysis of a subset of mouse liver immunocytes from the indicated group in (A). (D) Statistical difference analysis for immune cell subsets was obtained from dimensionality reduction analysis in panel (C). n = 5 male mice per group and data were presented as the mean ± SD. (E) Flow cytometry analysis of the ratio of the immune co-suppressive molecules (PD-1, TIM-3) and immune effector molecules (IFNγ, GzmB) positive cells in liver CD8+ T cells from the indicated group in (A). n = 6 male mice per group and data were presented as the mean ± SD. (F) Schema of coculture of human CD8+ T cells with HepG2 cells expressing Flag-EV or Flag-EPDR1. (G) Flow cytometry analysis of the ratio of immunosuppressive molecules (PD-1, TIM-3) and immune effector molecules (IFNγ, GzmB) positive cells in CD8+ T cells after coculture with the indicated tumor cells. n = 3 independent experiments and the data were presented as the mean ± SD. (H–J) Hepa 1-6 cells stably expressing Flag-EV or Flag-mEPDR1 were injected subcutaneously into C57BL/6 J mice (n = 6 male mice per group), and α-CD8 (4 mg/kg) neutralizing antibody was injected intraperitoneally four times (twice a week starting at 10 days after inoculation) to block CD8+ T cells and IgG2b was used as control. Tumor size was measured starting at 10 days after inoculation. Photographs show xenografts (H), growth curves (I), and relative tumor burdens (J) determined at the end of the experiment (day 25). Data were presented as the mean ± SEM. Data information: Statistical significance was determined by two-way ANOVA (D, I), one-way ANOVA (J), and two-tailed unpaired Student's t-test (B, E, G). Source data are available online for this figure.

of immune-related molecules in HepG2 cells. Our findings demonstrate a significant decrease in PD-L1 mRNA levels upon downregulation of EPDR1 compared to other investigated immune checkpoint molecules (Fig. 3A). Interestingly, treatment of HepG2 cells overexpressing EPDR1 or empty vector with actinomycin D resulted in comparable PD-L1 mRNA levels, suggesting that EPDR1 does not destabilize the PD-L1 mRNA molecule (Fig. EV3A). Furthermore, Western blotting and flow cytometry analyses indicated that EPDR1 promoted the expression of both global and membrane-bound PD-L1 (Figs. 3B and EV3B), whereas inhibiting EPDR1 had the opposite effect (Figs. 3C and EV3C). Upon extraction of tumor cells from the xenografts, we observed that EPDR1 knockdown in Hepa 1–6 cells significantly attenuated PD-L1 cell surface expression (Fig. EV3D). Considering the crucial role of PD-L1 in facilitating immune evasion by tumors, we hypothesize that EPDR1 modulates immune cell activity through the regulation of PD-L1 expression in tumor cells, potentially promoting immune evasion.

To unravel the mechanism underlying the regulation of PD-L1 by EPDR1, we conducted RNA sequencing (RNA-seq) analysis on both control and shEPDR1-treated HepG2 cells. Subsequent Kyoto Encyclopedia of Genes and Genomes (KEGG) pathway enrichment analysis revealed a significant downregulation of signaling pathways associated with NF-κB (Fig. 3D). Previous studies have reported that this pathway transcriptionally regulates PD-L1 expression and promotes the immune evasion of tumor cells (Maeda et al, 2018; Sun et al, 2018; Wang et al, 2019; Zhang et al, 2022). It implied that EPDR1 exerts its effect on PD-L1 expression through modulation of the NF-κB signaling pathway in cancer cells. We conducted a Western blotting analysis to confirm the decrease in p65 nuclear localization upon EPDR1 knockdown (Fig. 3E). P65 plays an important role as a transcription factor in the classical activation of the NF-κB pathway (Ghosh and Hayden, 2008). Treatment with TNFα induced activation of the NF-κB pathway, leading to p65 phosphorylation and subsequent upregulation of PD-L1 expression. In contrast, EPDR1 suppression attenuated this alteration (Fig. 3F). Additionally, overexpression of EPDR1 led to increased levels of PD-L1 expression and p65 phosphorylation, which were effectively inhibited by BAY11-7082, a potent NF-κB inhibitor (Fig. 3G). Similar results were obtained for PLC cells when treated with NF-κB agonists (TNFα) and inhibitors (BAY11-7082) (Fig. EV3E,F). Furthermore, we validated the interaction between p65 and the PD-L1 promoter through a chromatin

immunoprecipitation (ChIP) assay in HepG2 cells and observed a significant reduction upon EPDR1 knockdown (Fig. 3H).

To further investigate how EPDR1 regulates the mechanism underlying the modulation of the NF-κB signaling pathway, a proteomics analysis employing LC-MS/MS was performed and resulted in the identification of 110 proteins potentially binding to EPDR1 (Fig. 3I). Gene ontology analysis of biological processes and the abundance of potential binding proteins revealed that TRIM21 is the unique candidate as high-confidence interactor for EPDR1 which is involved in the NF-κB pathway (Fig. EV3G). We confirmed the binding of TRIM21 to EPDR1 in HepG2 and HEK293T cells through an immunoprecipitation assay (Figs. 3J and EV3H). The direct interaction between TRIM21 and EPDR1 was demonstrated through a pulldown assay (Fig. 3K). Confocal imaging revealed colocalization of the EPDR1 protein and TRIM21 in the cytosol (Fig. EV3I). Western blotting analysis confirmed that their respective protein levels were not affected by each other (Fig. EV3J). Previous studies have demonstrated that TRIM21 exerts its inhibitory effect on the NF-κB signaling pathway through ubiquitination and subsequent degradation of IKBKB (Gullà et al, 2018; Niida et al, 2010; Wada et al, 2009). We confirmed the interaction between TRIM21 and IKBKB by immunoprecipitation and pulldown assays (Fig. EV3K,L). Furthermore, we validated the negative regulatory role of TRIM21 in the IKBKB-NF-κB signaling pathway using Western blotting (Fig. EV3M). To elucidate the regulatory role of EPDR1 in the NF-κB signaling pathway through its interaction with TRIM21, we conducted a competitive binding experiment in HepG2 cells. Remarkably, we observed that overexpression of EPDR1 significantly inhibited the interaction between TRIM21 and IKBKB (Fig. 3L), which was further confirmed by the pulldown assay (Fig. 3M). Moreover, upon treatment with the autophagosome inhibitor 3-MA, overexpression of TRIM21 enhanced the ubiquitination of IKBKB, an effect that was subsequently attenuated by EPDR1 expression (Fig. 3N). The divergent regulatory effects of EPDR1 and TRIM21 on the IKBKB and NF-κB signaling pathways were further verified by Western blotting. Notably, our findings revealed that EPDR1 overexpression counteracted the inhibitory effect of TRIM21 on NF-κB (Fig. 3O). These findings convincingly suggest that EPDR1 binds to TRIM21, thereby impeding the interaction and degradation between TRIM21 and IKBKB. Consequently, this mechanism promotes the activation of the NF-κB pathway and subsequent upregulation of PD-L1 expression in cancer cells, ultimately facilitating tumor immune evasion.

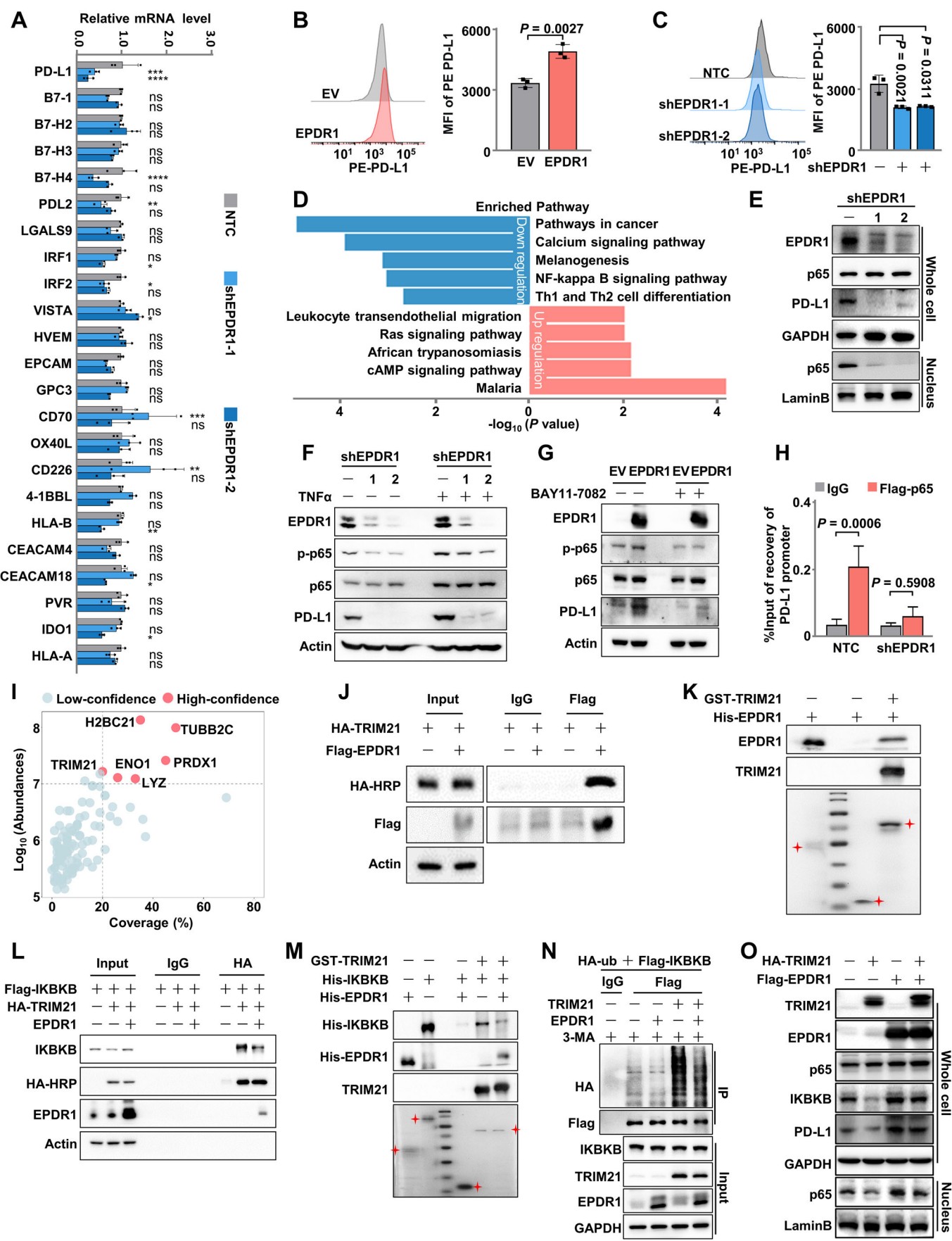

**Figure 3. EPDR1 enhances the NF-κB pathway and elevates PD-L1 expression in cancer cells by interacting with TRIM21.**

(A) qRT-PCR analysis of the mRNA levels of a series of immune-related molecules in HepG2 cells with EPDR1 knockdown. Data were presented as the mean ± SD of three independent experiments ($n = 3$). $**P \leq 0.01$, $***P \leq 0.001$, $****P \leq 0.0001$ compared with NTC control. (B) Flow cytometry analysis of the membrane-bound PD-L1 in HepG2 cells expressing Flag-EV and Flag-EPDR1. Data were presented as the mean ± SD of three independent experiments ($n = 3$). (C) Flow cytometry analysis of the membrane-bound PD-L1 in HepG2 cells with EPDR1 knockdown. Data were presented as the mean ± SD of three independent experiments ($n = 3$). (D) Histogram of the KEGG enrichment analysis results for differentially expressed genes between HepG2 cells expressing shEPDR1 and those expressing shNTC. $P$ value was computed using the one-sided Fisher's exact test and corrected for multiple hypothesis testing using a false discovery rate (FDR). (E) Western blotting analysis of the protein levels of PD-L1 and p65 in whole cell lysates and nuclear fractions in HepG2 cells expressing shNTC and shEPDR1 with GAPDH and Lamin B as loading controls, respectively. (F) Western blotting analysis of the protein levels of PD-L1 and p65 in HepG2 cells expressing shNTC and shEPDR1. Cells were treated with the NF-κB agonist TNFα (100 nM) or vehicle control 6 h before sample collection, and β-actin was used as a loading control. (G) Western blotting analysis of the protein levels of PD-L1 and p65 in HepG2 cells expressing Flag-EV and Flag-mEPDR1. Cells were treated with the NF-κB inhibitor BAY11-7082 (5 μM) or vehicle control 6 h before sample collection, and β-actin was used as a loading control. (H) ChIP analysis of the occupancy of p65 on the PD-L1 promoters in HepG2 cells expressing shNTC and shEPDR1. Data were presented as the mean ± SD of three independent experiments ($n = 3$). (I) Scatter diagram showing the potential interactors of EPDR1. (J) Co-IP assay showing the protein interaction between EPDR1 and TRIM21. HepG2 cells were infected with lentivirus carrying Flag-EV or Flag-EPDR1 together with HA-TRIM21 plasmids. Cell lysates were immunoprecipitated with an anti-Flag antibody, followed by Western blotting analysis with antibodies against Flag and HA tags. (K) Pull-down assay showing the protein interaction between GST-TRIM21 and His-EPDR1. GST-tagged TRIM21 and 6 × His-tagged EPDR1 proteins were purified from *E. coli* and incubated in vitro, followed by Western blotting analysis with antibodies against TRIM21 or EPDR1. The red asterisks indicate the target bands. (L) Co-IP assay showing the protein interaction between TRIM21 and IKBKB in the indicated genotypes. HepG2 cells expressing empty vector or EPDR1 were infected with lentivirus carrying Flag-IKBKB with HA-EV or HA-TRIM21 plasmids. Cell lysates were immunoprecipitated with an anti-HA antibody, followed by Western blotting analysis with antibodies against IKBKB, EPDR1, and HA tags. (M) Pull-down assay showing the protein interaction of GST-TRIM21 and His-EPDR1 or His-IKBKB. 6× His-tagged EPDR1 and IKBKB proteins were purified from *E. coli* and incubated with purified GST-tagged TRIM21 alone or together in vitro, followed by Western blotting analysis with antibodies against TRIM21, EPDR1 or IKBKB. The red asterisks indicate the target bands. (N) HEK293T cells expressing Flag-IKBKB and hemagglutinin-tagged ubiquitin (HA-Ub) were co-transfected with EV, EPDR1, TRIM21 plasmids alone, or EPDR1 plus TRIM21 for 48 h and treated with 10 mM 3-MA for 8 h before collection. Immunoprecipitation was performed using anti-Flag antibody or IgG using the above cells. Polyubiquitination of Flag-IKBKB was detected by Western blotting. (O) Western blotting analysis of the protein levels of PD-L1, IKBKB, and p65 in whole cell lysates and nuclear fractions in HepG2 cells expressing Flag-EPDR1 and/or HA-TRIM21, with GAPDH and Lamin B, respectively, as loading controls. Data information: Statistical significance was determined by two-way ANOVA (A, H), two-tailed unpaired Student's *t*-test (B), and one-way ANOVA (C). Source data are available online for this figure.

## The EPDR1-TRIM21-PD-L1 axis promotes tumor immune evasion in the HCC mouse model

To further demonstrate that PD-L1 is a functional EPDR1 target, we inoculated mEPDR1 or EV overexpressing Hepa 1–6 cells with PD-L1 knockdown into mice and detected tumor-infiltrating CD8+ T-cell exhaustion and activity markers. We found that mEPDR1 overexpression facilitated the in vivo tumor growth of Hepa 1–6 xenografts, which was largely abolished by PD-L1 knockdown (Fig. 4A–C). Further flow cytometric analysis revealed that mEPDR1 overexpression in cancer cells resulted in increased exhaustion and decreased activity of tumor-infiltrating T cells in mice, and this effect was abolished by PD-L1 knockdown (Fig. 4D,E). To eliminate other influences resulting from PD-L1 knockdown, we employed a xenograft model generated by subcutaneous or hepatic inoculation with murine liver cancer cell Hepa 1–6 overexpressing mEPDR1 or EV, and quantified the exhaustion and activity of tumor-infiltrating CD8+ T cells in tumor-bearing mice treated with or without anti-PD-L1 antibody (Fig. EV4A; Appendix Fig. S1A). Evaluation of tumor volume development, final weight, and tumor numbers revealed that the α-PD-L1 neutralizing antibody exhibited superior efficacy in suppressing tumor growth compared to the IgG control groups, particularly in cases where mEPDR1 was overexpressed (Fig. EV4B,C). After the isolation of immunocytes from the xenografts using density gradient centrifugation, flow cytometry assays were performed, revealing that α-PD-L1 neutralizing antibodies effectively suppressed the expression of the immunosuppressive molecules PD-1 and TIM-3 on CD8+ T cells (Fig. EV4D) while simultaneously restoring their cytotoxic function (Fig. EV4E). Similarly, in the hepatic portal vein xenograft model, high expression levels of mEPDR1 conferred a proliferative advantage to xenografts derived from Hepa 1–6 cells compared to controls. Treatment with α-PD-L1 neutralizing antibodies effectively suppressed tumor growth, particularly in the high mEPDR1 group. In comparison to the IgG2b control, α-PD-L1 neutralizing

antibodies resulted in a significant reduction in tumor amount (Appendix Fig. S1B). Characterization of CD8+ T cells in liver xenografts mirrored that observed in subdermal tumors (Appendix Fig. S1C,D). These results highlight the efficacy of neutralizing antibodies against tumors expressing EPDR1. Similarly, EPDR1 knockdown significantly attenuated tumor growth (Appendix Fig. S2A–C). Flow cytometric analysis showed that EPDR1 inhibition and anti-PD-L1 each resulted in significant suppression of exhaustion and increased antitumor activity in tumor-infiltrating CD8+ T cells. The infiltrated CD8+ T cells exhibited slightly, but not significantly, lower PD-1 and TIM-3 expression and higher IFNγ and GzmB expression in mice inoculated with EPDR1 knockdown Hepa 1–6 cells compared with those treated with anti-PD-L1 (Appendix Fig. S2D,E). The probable reason is that PD-L1 is not only expressed in cancer cells but also in tumor associated stromal cells. These data indicated that loss of EPDR1 in cancer cells enhanced the antitumor activity of tumor-infiltrating CD8+ T cells.

By employing a xenograft model generated by subcutaneous inoculation with Hepa 1–6 cells with differential expression levels of mEPDR1 and mTRIM21, we observed that mEPDR1 overexpression facilitated the in vivo tumor growth of Hepa 1–6 xenografts, which was largely abolished by mTRIM21 overexpression (Fig. 4F–H). We also found that mEPDR1 overexpression increased, while mTRIM21 overexpression decreased, the surface expression of PD-L1 in tumor cells (Fig. 4I). Further flow cytometric analysis showed that mTRIM21 expression could reverse the increased exhaustion and decreased activity of tumor-infiltrating T cells in mice inoculated with Hepa 1–6 cells overexpressing mEPDR1 (Fig. 4J,K). Moreover, we employed BAY11-7082 to inhibit NF-κB signaling in our xenograft model (Fig. EV4F). Assessment of tumor morphology, volume development, and final weight revealed that BAY11-7082 treatment prevented tumor progression, particularly in the group expressing

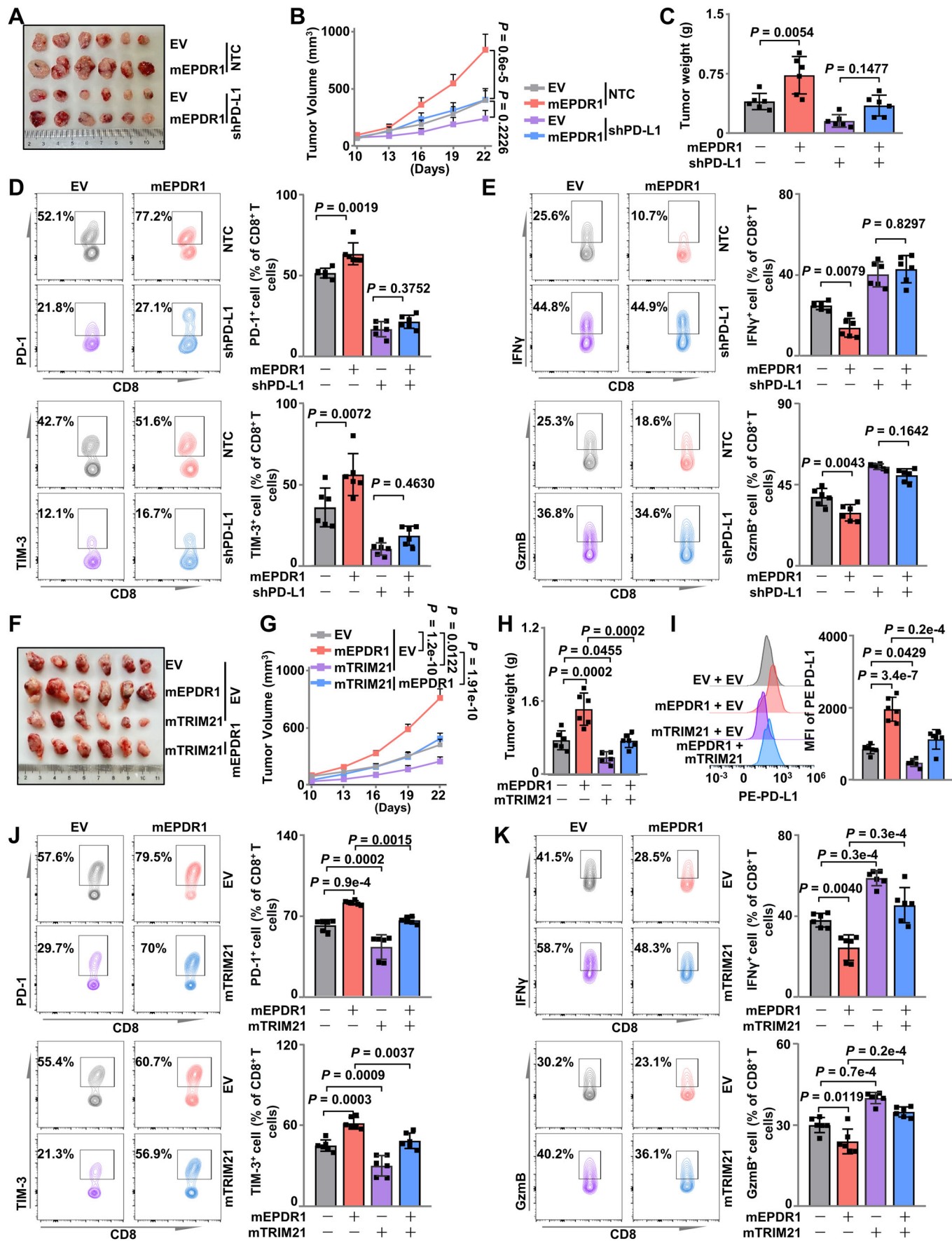

**Figure 4.   The EPDR1-TRIM21-PD-L1 axis promotes antitumor immune evasion in HCC.**

(A–C) Hepa 1-6 cells stably expressing Flag-EV and Flag-mEPDR1 were infected with shNTC or shPD-L1, and subsequently subcutaneously injected into C57BL/6J mice ($n = 6$ male mice per group). Tumor size was measured starting at 10 days after inoculation. Photographs show xenografts (A), growth curves (B), and tumor weight (C) determined at the end of the experiment (day 25). Data were presented as the mean ± SEM (B) and mean ± SD (C), respectively. (D) Flow cytometry analysis of the ratio of immunosuppressive molecules (PD-1, TIM-3) positive cells in tumor $CD8^+$ T cells from the indicated group in (A). Data were presented as the mean ± SD. (E) Flow cytometry analysis of the ratio of immune effector molecules (IFNγ, GzmB) positive cells in tumor $CD8^+$ T cells from the indicated groups in (A). Data were presented as the mean ± SD. (F–H) Hepa 1-6 cells stably expressing Flag-EV, Flag-mTRIM21, Flag-mEPDR1, or Flag-mTRIM21 plus Flag-mEPDR1 both were injected subcutaneously into C57BL/6 J mice ($n = 6$ male mice per group). Tumor size was measured starting at 10 days after inoculation. Photographs show xenografts (A), growth curves (B), and tumor weight (C) determined at the end of the experiment (day 25). Data were presented as the mean ± SEM (G), and mean ± SD (H), respectively. (I) Flow cytometry analysis of membrane-bound PD-L1 in the indicated group in (A). Data were presented as the mean ± SD. (J) Flow cytometry analysis of the ratio of immunosuppressive molecules (PD-1, TIM-3) positive cells in tumor $CD8^+$ T cells from the indicated group in (A). Data were presented as the mean ± SD ($n = 6$). (K) Flow cytometry analysis of the ratio of immune effector molecules (IFNγ, GzmB) positive cells in tumor $CD8^+$ T cells from the indicated group in (A). Data were presented as the mean ± SD ($n = 6$). Data information: Statistical significance was determined by two-way ANOVA (B, G) and one-way ANOVA (C–E, H–K). Source data are available online for this figure.

mEPDR1 (Fig. EV4G,H). Furthermore, the exhaustion state of $CD8^+$ T cells (Fig. EV4I) and their impaired functionality (Fig. EV4J), induced by high mEPDR1 expression, were restored upon administration of BAY11-7082. These results confirm the significance of the EPDR1-TRIM21-PD-L1 axis in promoting immune evasion and tumorigenesis.

## The level of EPDR1 expression is positively correlated with p65 nuclear location and PD-L1 transcription in clinical HCC tissues

To assess whether the signaling axis discovered in this study is operational in human liver cancer, we conducted an examination of these proteins in clinical HCC samples (Fig. 5A). Upon quantifying the respective staining intensity, a moderate positive correlation of EPDR1 with PD-L1 as well as nuclear p65 was revealed (Fig. 5B,C) Furthermore, we observed a positive correlation between EPDR1 and PD-L1 expression at mRNA level (Fig. EV5A). Analysis of samples from the liver cancer mouse model confirmed the positive correlation of EPDR1 and PD-L1 at the protein level (Fig. EV5B).

In conclusion, our study identified an oncogene, EPDR1, which attenuates the degradation of IKBKB by competing for binding with TRIM21, leading to increased NF-κB signaling and PD-L1 expression. This results in $CD8^+$ T-cell exhaustion and tumor development (Fig. 5D). Our findings have contributed to the understanding of PD-L1 regulation, proposing a novel target for the treatment of HCC.

## Discussion

Although immune checkpoint blockade (ICB) represents a more efficacious strategy than traditional regimens in some cancer patients, identifying the patients who would benefit from this treatment remains a significant challenge (Chan et al, 2019; Morad et al, 2021). In this study, we set out to search for the genes that affect checkpoint molecule expression and ICB treatment efficiency by examining the gene expression pattern of liver cancer patients who may benefit from ICB treatment and identified the oncogene EPDR1 as a potential candidate gene associated with a positive response to ICB treatment. Although EPDR1 has been studied in colorectal cancer and is considered a protumor molecule capable of accelerating tumor metastasis (Chu et al, 2018), its function remains controversial, and the specific mechanism underlying its actions is yet to be fully elucidated. Our findings revealed

the protumor role of EPDR1 in liver cancer and elucidated its modulation of PD-L1 and antitumor immunity for the first time. Our results enhance the understanding of how the tumor microenvironment is influenced by a tumor intrinsic regulator and provide a new perspective on PD-L1 regulation and ICB treatment.

Intriguingly, in this study, we discovered that EPDR1 exerts its antitumor immune function by disrupting the interaction between the E3 ubiquitin ligase TRIM21 and IKBKB, which subsequently regulates the exhaustion of antitumor $CD8^+$ T cells. This suggests the potential role of TRIM21 in modulating tumor immunity. Previous studies have suggested that ubiquitin ligases serve as central regulatory nodes of signaling pathways and may become favorable targets for cancer intervention because they may be mutated/dysfunctional in tumor cells and target specific substrates in a spatial and temporal manner (Senft et al, 2018; Zhou and Sun, 2021). While TRIM21 has been reported to be involved in various important cellular activities in cancer cells, such as cell metabolism, cytokinesis, and redox regulation, it also contributes to the inhibition of proinflammatory activity in immune cells (Alomari, 2021; Li et al, 2020; Lu et al, 2023). Its intricate regulatory role in the NF-κB pathway has been revealed, demonstrating both positive or negative function by binding to multiple upstream and downstream members of this signaling cascade (Huang et al, 2024). High expression levels of TRIM21 appear to enhance the activity of the NF-κB pathway in immune cells (McEwan et al, 2013; Xiao et al, 2021). However, contrasting effects have been documented in other cell types (Gullà et al, 2018; Niida et al, 2010; Wada et al, 2009; Yoshimi et al, 2009). Notably, the impact of TRIM21 on tumor immunity also exhibits a dual nature, as it can either promote or inhibit antitumor immune responses by directly binding to and destabilizing its substrates (Du et al, 2023; Fu et al, 2023; Li et al, 2023; Shi et al, 2022), including PD-L1 (Gao et al, 2021; Sun et al, 2023). These seemingly contradictory observations limit the utility of TRIM21 as a potential target for tumor therapy. In this regard, our findings suggest that targeting EPDR1 may be an effective strategy for the treatment of tumors with dysfunctional TRIM21.

In conclusion, our findings reveal that EPDR1 is capable of modulating the nuclear translocation of p65 and subsequent transactivation of PD-L1 via a new TRIM21-mediated mechanism. These processes ultimately lead to the exhaustion of $CD8^+$ T cells and tumor immune evasion. Our study significantly contributes to the understanding of PD-L1 regulation and identifies EPDR1 as a potential target for combination therapy with PD-L1 inhibitors. Thus, targeting EPDR1 and TRIM21 holds promise as a novel strategy for improving liver cancer immunotherapy.

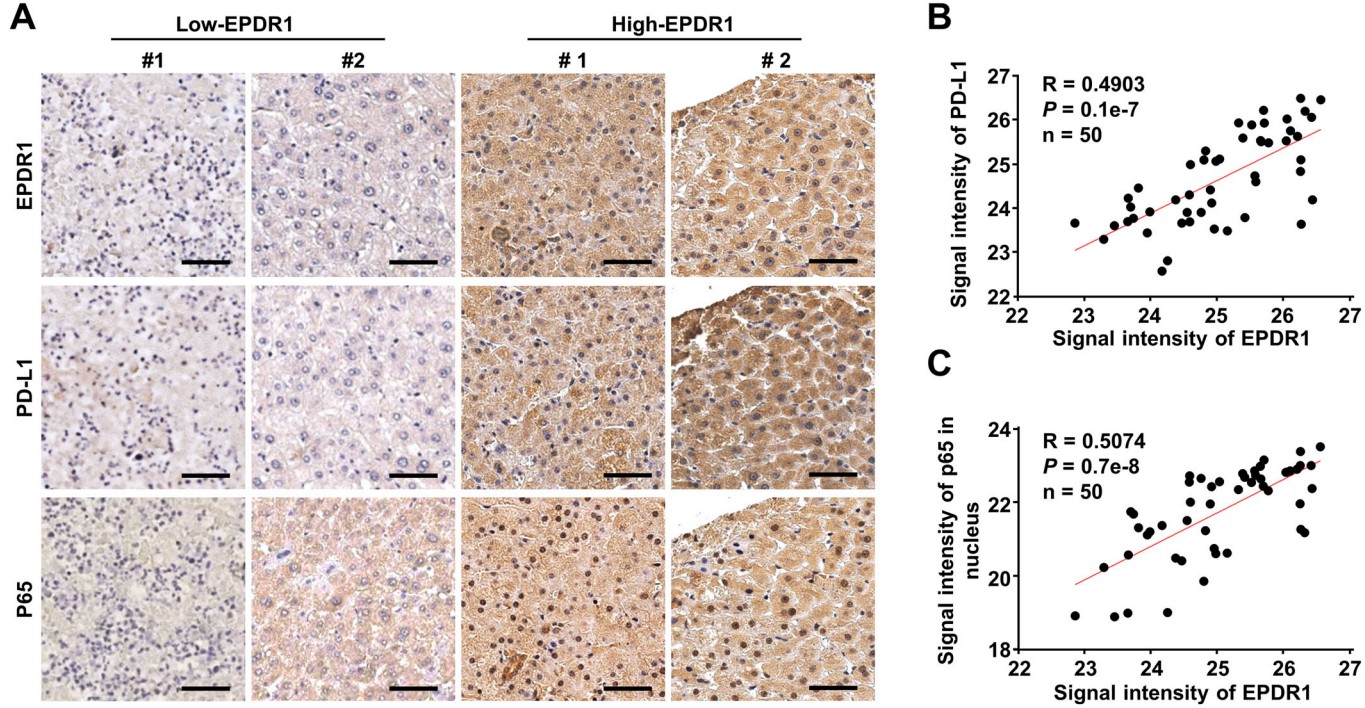

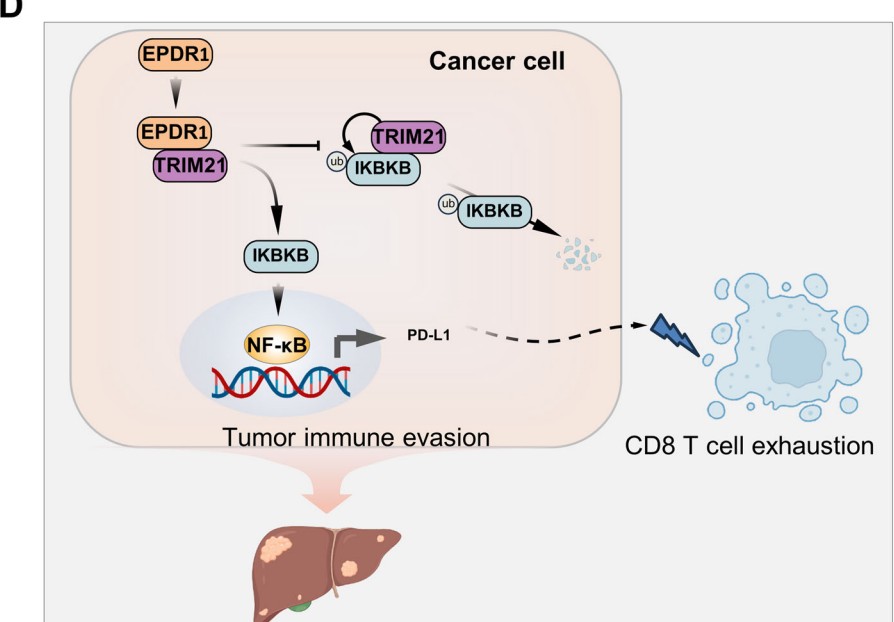

**Figure 5. The level of EPDR1 expression is positively correlated with p65 nuclear location and PD-L1 in clinical HCC tissues.**

(A) Representative immunohistochemistry images of EPDR1, PD-L1, p65, and CD8 staining in HCC specimens; scale bars, 50 μm. (B) Correlation analysis of EPDR1 and PD-L1 positive signal in HCC specimens, *P* values and R were calculated by two-tailed Person's correlation analysis. (C) Correlation analysis of EPDR1 and nucleus located p65 positive signal in HCC specimens, *P* values and R were calculated by two-tailed Person's correlation analysis. (D) Summary: In the presence of EPDR1, intracellular EPDR1 directly binds to TRIM21 in competition with IKBKB, thereby disrupting the ubiquitylation and degradation of IKBKB, leading to enhanced NF-κB signaling and downstream PD-L1 expression, which exacerbates the exhaustion of CD8[+] T cells and contributes to tumor progression. Source data are available online for this figure.

# Methods

**Reagents and tools table**

| Reagent/resource | Reference or source | Identifier or catalog number |
| --- | --- | --- |
| **Experimental Models** | | |
| HepG2 cells (H. sapiens) | ATCC | HB-8065 |
| PLC cells (H. sapiens) | ATCC | CRL-8024 |
| Hepa 1–6 (M. musculus) | ATCC | CRL-1830 |
| HEK293T (H. sapiens) | Wuhan Pricella Biotechnology | CL-0005 |
| BALB/c (M. musculus) | Shanghai Laboratory Animal Center | N/A |
| C57BL/6 J (M. musculus) | Cyagen Biosciences | N/A |
| **Recombinant DNA** | | |
| pSin-Flag-EPDR1 | This study | |
| pSin-EPDR1 | This study | |
| pSin-HA-TRIM21 | This study | |
| pSin-Flag-TRIM21 | This study | |
| pSin-TRIM21 | This study | |
| pSin-Flag-IKBKB | This study | |
| pSin-HA-IKBKB | This study | |
| pET-32a-His-EPDR1 | This study | |
| pET-32a-His-IKBKB | This study | |
| Pgex-4T-1-GST-TRIM21 | This study | |
| pSin-HA-ubiquitin | This study | |
| pSin-Flag-mEPDR1 | This study | |
| pSin-HA-mTRIM21 | This study | |
| pBl-Myc-EPDR1 | This study | |
| **Antibodies** | | |
| Mouse anti-β-Actin | Proteintech | 60008-1-Ig |
| Rabbit anti-EPDR1 | Thermo Fisher Scientific | PA5-50404 |
| Mouse anti-α-Tubulin | Proteintech | 66031-1-Ig |
| Rabbit anti-TRIM21 | Cell Signaling Technology | Cat# 92043 S |
| Rabbit anti-p65 | Cell Signaling Technology | Cat# 8242 T |
| Rabbit anti-Phospho-p65 | Cell Signaling Technology | Cat# 3003 T |
| Rabbit anti-IKBKB | Proteintech | 15649-1-AP |
| Mouse anti-GAPDH | Proteintech | 60004-1-Ig |
| Rabbit anti-Lamin B | Abcam | ab176880 |
| Rabbit anti-PD-L1 | Proteintech | 28076-1-AP |
| Rabbit anti-PD-1 | Abcam | AB216352 |
| Mouse anti-CD8 | Cell Signaling Technology | Cat #70306 |
| Mouse anti-Flag-M2 | Sigma | F1804 |
| Mouse anti-HA | Proteintech | 51064-2-AP |
| Rabbit anti-Calnexin | Proteintech | 10427-2-AP |
| HRP-conjugated anti-rabbit | Bio-Rad | Cat# 170-6515 |
| HRP-conjugated anti-mouse | Bio-Rad | Cat# 170-6516 |
| CoraLite488-conjugated Goat Anti-Mouse IgG(H + L) | Proteintech | Cat# SA00013-1 |
| CoraLite594-conjugated Goat Anti-Mouse IgG(H + L) | Proteintech | Cat# SA00013-4 |

| Reagent/resource | Reference or source | Identifier or catalog number |
|---|---|---|
| Purified anti-mouse CD3ε | Biolegend | Cat# 100302 |
| Purified anti-mouse CD28 | Biolegend | Cat# 122002 |
| Anti-mouse CD16/32 | BD Biosciences | Cat# 553142 |
| Mus CD45.2 | Biolegend | 104 |
| Mus F4/80 | Biolegend | BM8 |
| Mus CD11b | BD Horizon | M1/70 |
| Mus Gr-1 | BD Horizon | RB6-8C5 |
| Mus MHC II | Biolegend | M5/114.15.2 |
| Mus NK1.1 | Biolegend | PK136 |
| Mus CD3e | Biolegend | 17A2 |
| Mus CD4 | Biolegend | GK1.5 |
| Mus CD8 | Biolegend | 54-6.7 |
| Mus PD-1 | Biolegend | 29 F.1A12 |
| Mus CD19 | Biolegend | 6D5 |
| Mus CD3e | Biolegend | 500A2 |
| Mus CD4 | Biolegend | RM4-5 |
| Mus IFN-γ | Biolegend | XMG1.2 |
| Mus TNF-α | Biolegend | MP6-XT22 |
| Mus Granzyme B | Biolegend | QA16A02 |
| Mus Tim3 | Biolegend | RMT3-23 |
| Mus LAG3 | Biolegend | C9B7W |
| Mus PD-L1 | Biolegend | 29E.2A3 |
| Homo PD-L1 | Biolegend | 10 F.9G2 |
| Homo CD45 | Biolegend | 2D1 |
| Homo CD3 | Biolegend | OKT3 |
| Homo CD8a | Biolegend | HIT8a |
| Homo PD-1 (CD279) | Biolegend | NAT105 |
| Homo TIM-3 (CD366) | Biolegend | A18087E |
| Homo LAG3 (CD223) | Biolegend | 11C3C65 |
| Homo IFN-γ | Biolegend | W19227A |
| Homo TNF-α | Biolegend | MAb11 |
| Homo Granzyme B | Biolegend | QA18A28 |
| **Oligonucleotides and other sequence-based reagents** | | |
| Non-target control | - | CCGGCAACAAGATGAAGAGCACCAACTCGAGTTGGTGCTCTTCATCTTGTTGTTTTT |
| shEPDR1-1 | Homo sapiens | CCGGCCTGCAAGAGATTATTTGAATCTCGAGATTCAAATAATCTCTTGCAGGTTTTTG |
| shEPDR1-2 | Homo sapiens | CCGGGATCCTCTTGACATTCCTCAACTCGAGTTGAGGAATGTCAAGAGGATCTTTTTG |
| shTRIM21-1 | Homo sapiens | CCGGACCCTCTGTCCACTGAATATTCTCGAGAATATTCAGTGGACAGAGGGTTTTTTG |
| shTRIM21-2 | Homo sapiens | CCGGCTGCTGCAGGAGGTGATAATTCTCGAGAATTATCACCTCCTGCAGCAGTTTTTG |

| Reagent/resource | Reference or source | Identifier or catalog number |
|---|---|---|
| shEPDR1-1 | Mus musculus | CCGGTAAGAATCGAAGGTGATA ATACTCGAGTATTATCACCTTC GATTCTTATTTTTG |
| shEPDR1-2 | Mus musculus | CCGGCTTGGACAATCTTAGTAT TTACTCGAGTAAATACTAAGAT TGTCCAAGTTTTTG |
| shPD-L1 | Mus musculus | CCGGGACGTCAAGCTGCAGGAC GCCTCGAGGCGTCCTGCAGCTT GACGTCTTTTTG |
| Homo 18 s | CGCTACTACCGATTGGATGG | AGTTCGACCGTCTTCTCAGC |
| Homo B7-1 | GGCCCGAGTACAAGAACCG | TCGTATGTGCCCTCGTCAGAT |
| Homo B7-H2 | GCAGCCTTCGAGCTGATACTC | GTTTTCGACTCACTGGTTTGC |
| Homo B7-H3 | CTGGCTTTCGTGTGCTGGAGAA | GCTGTCAGAGTGTTTCAGAGGC |
| Homo B7-H4 | CTCACAGATGCTGGCACCTACA | GCAAGGTCTCTGAGCTGGCATT |
| Homo PDL2 | ACCCTGGAATGCAACTTTGAC | AAGTGGCTCTTTCACGGTGTG |
| Homo LGALS9 | ACACCCAGATCGACAACTCCTG | CAAACAGGTGCTGACCATCCAC |
| Homo IRF1 | GAGGAGGTGAAAGACCAGAGCA | TAGCATCTCGGCTGGACTTCGA |
| Homo IRF2 | TAGAGGTGACCACTGAGAGCGA | CTCTTCATCGCTGGGCACACTA |
| Homo VISTA | ACGCCGTATTCCCTGTATGTC | TTGTAGAAGGTCACATCGTGC |
| Homo HVEM | GTGCAGTCCAGGTTATCGTGT | CACTTGCTTAGGCCATTGAGG |
| Homo EPCAM | ATAACCTGCTCTGAGCGAGTG | ATAACCTGCTCTGAGCGAGTG |
| Homo GPC3 | ATTGGCAAGTTATGTGCCCAT | TTCGGCTGGATAAGGTTTCTTC |
| Homo CD70 | GCTTTGGTCCCATTGGTCG | CGTCCCACCCAAGTGACTC |
| Homo OX40L | CCAGGCCAAGATTCGAGAGG | CCGATGTGATACCTGAAGAGCA |
| Homo CD226 | ATAGCCACATTGTTTCGGAACC | ATCTGACGGGGCTGGATCTTT |
| Homo 4-1BBL | GGCTGGAGTCTACTATGTCTTCT | ACCTCGGTGAAGGGAGTCC |
| Homo HLA-B | CAGTTCGTGAGGTTCGACAG | CAGCCGTACATGCTCTGGA |
| Homo CEACAM4 | ATTCAAGCAAATATCCCAGGGG | GGCATTTATGGTTCGTAGGGTG |
| Homo CEACAM18 | GCCTACCTCTAGTAGTGACCG | CTGGGAAACTCTCTATCATGCAC |
| Homo PVR | TGGAGGTGACGCATGTGTC | GTTTGGACTCCGAATAGCTGG |
| Homo IDO1 | CCTGATCTCATAGAGTCTGGC | TGCATCCCAGAACTAGACGTGC |
| Homo PD-L1 | TGCCGACTACAAGCGAATTACTG | CTGCTTGTCCAGATGACTTCGG |
| Homo HLA-A | ACCCTCGTCCTGCTACTCTC | CTGTCTCCTCGTCCCAATACT |
| Homo EPDR1 | GTCCAGGAGTGGTCGGACA | CCGCGTAGACAATATCACACTG |
| Mus 18 s | GGCCGTTCTTAGTTGGTGGAGCG | CTGAACGCCACTTGTCCCTC |
| Mus EPDR1 | ACATCTGACTGTGGCTGCTC | AGCGGTCAGTACTCTTGCAC |
| Homo CHIP control | TTGCAAAATCACATTTTCTTTCTGGAAAT | AAACTCTGGGATCTCCCAGGG |
| Homo PD-L1 promoter | AAAGCCATATGGGTCTGCTG | CAACATCTGAACGCACCTTG |
| **Chemicals, enzymes and other reagents** | | |
| Fetal Bovine Serum | Biological Industries | 04-001-1ACS |
| Medium Dulbecco's Modified Eagle Medium | Thermo Fisher Scientific | 12800-017 |
| HEPEs | Thermo Fisher Scientific | 15630-080 |
| Penicillin-Streptomycin Solution | Biological Industries | 03-031-1B |
| 0.25% Trypsin | Biological Industries | B03-050-1A |
| Opti-MEM | Thermo Fisher Scientific | 31985-088 |
| Recombinant Human TNFα | Proteintech | HZ-1014 |
| InVivoMAb anti-mouse PD-L1 | BioXcell | BE0101 |

| Reagent/resource | Reference or source | Identifier or catalog number |
|---|---|---|
| InVivoMAb anti-mouse CD8 | BioXcell | BE0061 |
| InVivoMAb rat IgG2b isotype control | BioXcell | BE0090 |
| Mouse anti-human PD-L1 | BioXcell | BE0383 |
| Mouse IgG control | BioXcell | BE0083 |
| BAY11-7082 | MCE | HY-13453 |
| Actinomycin D | MCE | HY-17559 |
| 3-MA | MCE | HY-19312 |
| Puromycin | Sigma-Aldrich | P8833-100 mg |
| Polybrene | Sigma-Aldrich | H9268-5 G |
| Dimethyl Sulfoxide | Sangon Biotech | A100231-0500 |
| Cocktail | Sigma-Aldrich | 5056489001 |
| PMSF | Sangon Biotech | A610425-0005 |
| Protein A/G Beads | Thermo Fisher Scientific | 53133 |
| Trizol | Thermo Fisher Scientific | 15596-018 |
| T4 ligase | Thermo Fisher Scientific | 15224017 |
| CD8(TIL) MicroBeads | Miltenyi Biotech | 130-116-478 |
| Cell Stimulation cocktail (500 ×) | Thermo Fisher | 00-4970-93 |
| Collagenase IV | Sigma-Aldrich | C5138-1G |
| Ligation-Free Cloning System | abmGood | E001-5-B |
| Murine IL-2 | Proteintech | Cat# 212-12 |
| Human IL-2 | PeproTech | Cat# AF-200-02-50 |
| Ampicillin sodium | Sangon Biotech | A610028-0025 |
| HiScript II 1st Strand cDNA Synthesis Kit | Vazyme | R211-02 |
| AceQ® qRT-PCR, SYBR® Green Master Mix | Vazyme | Q111-03 |
| 2 × Taq Master Mix(Dye) | Vazyme | P112-03 |
| Percoll | Cytiva | 17089109 |
| 2-Mercaptoethanol | Gibco | Cat# 21985-023 |
| CD8 MicroBeads, human | Miltenyi Biotech | Cat# 130-045-201 |
| Dynabeads™ Human T-Activator CD3/CD28 | Miltenyi Biotech | Cat# 130-091-441 |
| Paraformaldehyde-glutaraldehyde | Leagene Biotechnology | Cat# DF0139 |
| **Software** | | |
| GraphPad Prism 8 | https://www.graphpad.com/ | |
| Flow Jo 10.0 | https://www.flowjo.com/ | |
| **Other** | | |

## Cell culture and reagents

Human liver cancer cell lines (HepG2, PLC), murine liver cancer cell lines (Hepa 1–6), and human renal epithelial cell lines (HEK293T) were cultured in DMEM (Gibco, USA) supplemented with 10% FBS (BI, Israel) and 1% penicillin-streptomycin (Invitrogen, USA). CD8+ T cells were cultured in RPMI-1640 (Gibco) with 10% FBS, 20 ng/ml IL-2, and 1% Pen/Strep after activation. All cell lines were tested for mycoplasma contamination and no cell lines were contaminated. All cells were grown at 37 °C and 5% $CO_2$. All drugs used in this study are listed in the Reagents and Tools Table.

## Plasmids and established stable cells

All shRNAs in the PLKO.1 vector against EPDR1 and TRIM21 were obtained commercially (Sigma-Aldrich). shRNA targeting sequences are listed in the Reagents and Tools Table. EPDR1, TRIM21, and IKBKB were subcloned and inserted into the pSin-3×Flag or pSin-HA empty vector; they were then co-transfected with plasmids encoding VSVG and Δ8.9 into HEK293T packaging cells using PEI (Polysciences). HepG2, PLC or Hepa 1–6 cells were infected with lentivirus containing polybrene and selected with 0.5 μg.ml⁻¹ puromycin to establish stable cells.

## Western blotting assay

Cultured cells were lysed in RIPA buffer (50 mM Tris-HCl (pH 8.0), 150 mM NaCl, 5 mM EDTA, 0.1% SDS, and 1% NP-40) supplemented with protease inhibitor cocktails (Roche, Switzerland) and 100 μM phenylmethylsulfonyl fluoride (PMSF). The protein concentration in the lysate was quantified using the Bradford method (Sangon Biotech, China). Equal amounts of protein lysate were boiled and fractionated by 7–11% SDS–PAGE. Signals were detected using Western ECL Substrate (Bio-Rad). All primary antibodies used for immunoblotting are listed in the Reagents and Tools Table.

## qRT–PCR

According to the manufacturer's instructions, total RNA was extracted from cells or tissues using TRIzol (Life Technologies), and complementary DNA was synthesized from 1–3 μg of RNA using the iScript cDNA Synthesis Kit (Bio-Rad). qRT-PCR was performed using SYBR Green Master Mix (Vazyme). Primer sequences are shown in the Reagents and Tools Table. All samples were normalized to housekeeping genes (*Actin* or *18s*).

## mRNA stability assay

HepG2 cells incubated with complete DMEM were treated with 5 μM actinomycin D for 0, 5, or 10 h. No decrease in cell viability was observed during the experiment. Total RNA was collected with TRIzol, and mRNA levels were analyzed by qRT-PCR. For all samples, the target mRNA levels were normalized to those of *18s*.

## Immunoprecipitation

HEK293T or HepG2 cells were lysed with IP buffer (1% NP-40, 20 mM Tris-HCl pH 7.5, 150 mM NaCl, 2 mM EDTA, 1.5 mM MgCl$_2$) supplemented with protease inhibitor cocktail for 2 h on ice and centrifuged at $12,000 \times g$ for 10 min at 4 °C. The supernatant was incubated with the designated primary antibody at 4 °C overnight, and then protein A/G-conjugated beads were incubated for an additional 1 h. The immune precipitates were washed three times with 0.5% NP-40 IP buffer and then boiled with SDS buffer, followed by immunoblot analysis.

## Pull-down assay

The cDNAs encoding EPDR1 or IKBKB were cloned and inserted into the pET-32a vector (Novagen), and cDNA encoding TRIM21 was cloned and inserted into the Pgex-4T-1 vector (GE Healthcare) by the 5× Ligation-Free Cloning Master Mix (ABM). The target proteins were induced with 1 mM IPTG for 18 h at 16 °C. Purified His-tagged proteins and GST-fused proteins were incubated in pull-down buffer (137 mM NaCl, 2.7 mM KCl, 10 mM Na$_2$HPO$_4$, 2 mM KH$_2$PO$_4$, 0.02 mM EDTA, 0.1% Triton X-100). After incubation, the beads were pelleted and washed three times with a pulldown buffer. Protein samples were analyzed by Western blotting.

## RNA sequencing and data analysis

Total RNA was extracted from cell lines using TRIzol Reagent (Life Technologies). RNA integrity was assessed by RNA integrity number and determined using an Agilent 2100 Bioanalyzer. A total of 3 μg of RNA per sample was used for analysis. Sequence sampling was performed from one single replicate. Libraries were generated using a NEBNext Ultra RNA Library Prep Kit for Illumina (NEB). RNA-seq was performed on an Illumina NovaSeq 6000 platform by Novogene (Tianjin). Reads were aligned to the human genome hg19. TopHat2 v.2.1.0 and cufflinks v.2.2.1 were used to analyze RNA-seq data. Gene differential expression analysis was carried out with the DEGSeq R package (1.26.0). Gene set enrichment analysis was performed via the DAVID Bioinformatics Resources.

## Ubiquitination assay

HEK293T cells were co-transfected with pSin-HA-ubiquitin and pSin-Flag-IKBKB in the presence of either pSin-EPDR1, pSin-TRIM21, or both, as indicated. After incubation for 48 h, the autophagosome inhibitor 3-MA was added to the culture medium for an additional 12 h, followed by collection of the cells and protein lysis with SDS buffer. Equal amounts of protein lysates were immunoprecipitated with an anti-Flag-M2 antibody and subjected to SDS–PAGE, followed by blotting with an anti-ubiquitin antibody.

## ChIP-qRT-PCR assay

The ChIP assay was performed with the EZ-ChIP Kit (Millipore) following the manufacturer's instructions. Briefly, cells were fixed with 1% formaldehyde and quenched with 0.125 M glycine. Cells were sonicated using an Ultrasonic Cell Disruptor (Scientz). DNA was immunoprecipitated with either control IgG or Flag-M2 primary antibody. RNA and protein were digested using RNase A and proteinase K, respectively, followed by qRT-PCR analysis. The primers used for analysis are listed in the Reagents and Tools Table.

## In vivo depletion of CD8$^+$ T cells

To deplete CD8$^+$ T cells in vivo, mice were injected intraperitoneally with 150 μg of anti-CD8 antibody (α-CD8) 3 days before tumor inoculation and twice weekly thereafter to ensure sustained depletion of the CD8$^+$ T-cell subset during the experimental period. In contrast, groups of mice were treated with IgG2b isotype control.

## Coculture experiment

Mouse CD8$^+$ T cells were separated from C57BL/6 mouse spleens with a CD8$^+$ T-cell enrichment kit. CD8$^+$ T cells were stimulated with plate-bound anti-CD3 at the indicated concentration and in fully supplemented tissue culture medium (RPMI plus 10% FBS, 1 μg/mL anti-CD28, 25 mM HEPES, 5 μM mercaptoethanol, 1% penicillin‑streptomycin, 10 μg/mL IL-2) for 3 days. Hepa 1–6 cells stably transfected with the indicated vector were incubated with activated mouse CD8$^+$ T cells for 3 days. Samples were analyzed by flow cytometry.

Human CD8$^+$ T cells were separated from PBMCs with a CD8$^+$ T-cell enrichment kit. CD8$^+$ T cells were cultured in RPMI-1640 medium and activated with Dynabeads™ Human T-Activator CD3/CD28 for 1 week according to the manufacturer's instructions. HepG2 cells with the indicated vector were seeded into 24-well

plates and incubated with activated mouse CD8$^+$ T cells for 3 days. Samples were analyzed by flow cytometry.

## Flow cytometry

Single-cell suspensions were prepared from cells in culture or tumors of HCC-bearing mice. For tumor samples from HCC-bearing mice, a single-cell suspension was obtained by rapid and gentle stripping, physical grinding, and filter filtration. Single lymphocytes were blocked with CD16/CD32 antibody and stained with the indicated fluorochrome-conjugated antibodies for 30 min at 4 °C. For intracellular cytokine staining, cells were incubated with a stimulation cocktail for 4 h prior to cell surface and cytokine staining. Then, cell staining was performed after fixation and permeabilization with antibodies against the murine samples. Then, stained cells were analyzed by BD FACSAria SORP or BD FACSCelestaTM flow cytometer. All analyses of flow cytometry data were performed using Flow Jo 10.0 software. The antibodies used in this study are listed in the Reagents and Tools Table.

## Immunofluorescence staining

Cells were fixed with paraformaldehyde-glutaraldehyde at room temperature for 30 min. Then, the membrane was broken with 1% Triton X-100 for 30 min, followed by blocking with 5% BSA. The primary antibody was incubated at 4 °C overnight, and the secondary antibody was incubated at room temperature for 1 h. Finally, an anti-fluorescence quenching sealing solution (including DAPI) was used for sealing. Images of IF staining were captured using a Nikon Ti-ea1 laser scanning confocal microscope (Nikon), and data were analyzed using NIS-Elements Viewer microscope imaging software. Primary antibodies or reagents against the targeted proteins were used: HA-Tag and EPDR1. Anti-rabbit or anti-mouse secondary antibodies conjugated to CoraLite488 or CoraLite594 were used.

## Animal studies

All animals were housed at a suitable temperature (22–24 °C) and humidity (40–70%) under a 12/12-h light/dark cycle with unrestricted access to food and water for the duration of the experiment. All animal studies were approved by the Animal Research Ethics Committee of the University of Science and Technology of China. For xenograft experiments, $2 \times 10^6$ or $3 \times 10^5$ Hepa 1–6 cells with indicated genotypes were subcutaneously or hepatoportal injected into 5-week-old male mice, respectively (BALB/c nude mice; Shanghai Laboratory Animal Center. C57BL/6J mice; Cyagen Biosciences, Inc.). The following formula was used to calculate tumor volume: length × width$^2$ × 0.52. The xenograft tumor burden was less than the maximum tumor size (2 cm³) approved by the Animal Research Ethics Committee of the University of Science and Technology of China. YAP5SA-induced HCC model was built with 4-week-old male C57BL/6J mice (Shanghai SLAC Laboratory Animal Co.) as described in our previous study (Wang et al, 2023). Briefly, plasmids expressing human YAP5SA plus RFP or human YAP5SA plus mouse-EPDR1, along with PB transposase-expressing plasmids, were diluted in sterile Ringer's solution to a volume equal to 10% of body weight and injected via the tail vein within 5–7 s. At the end of animal studies, all mice were euthanized by inhaling carbon dioxide.

## Clinical human HCC specimens

Snap-frozen HCC tissues and corresponding noncancerous tissues that were at least 2 cm distant from the edge of tumors were collected from 30 patients with HCC in the First Affiliated Hospital of the University of Science and Technology of China. Total RNA and protein were extracted from paired HCC and noncancerous tissues and then measured by qRT-PCR and immunoblotting, respectively. Formalin-fixed, paraffin-embedded HCC tissues and adjacent non-cancerous tissues were collected from patients with HCC in the First Affiliated Hospital of the University of Science and Technology of China. Patients provided written informed consent to use these clinical materials for research purposes, and the study was approved by the Institutional Research Ethics Committee of the First Affiliated Hospital of the University of Science and Technology of China. All patients volunteered and received no compensation.

## IHC

The IHC procedure was performed according to standard protocols. In brief, Paraffin-embedded tissues were sliced into 4-μm thick sections for hematoxylin and eosin (H&E) or IHC staining. The sections were incubated with anti-EPDR1 (Thermo Fisher, 1:200), anti-PD-L1 (Proteintech, 1:500) or anti-p65 (Cell Signaling Technology, 1:200) antibodies overnight at 4 °C. After incubation with HRP-conjugated secondary antibodies, the sections were visualized with DAB. Images were randomly obtained at 200× magnification using Leica AperioCS2, and quantification was performed with Image pro plus software.

## LC-MS and data analysis

The sample was prepared according to the instructions of a preparation kit (Applied Protein Technology, Shanghai, China). LC-MS and data analysis were performed by Applied Protein Technology (Shanghai, China). The immunoprecipitation-mass spectrometry data of protein binding with EPDR1 in HepG2 cells are provided in the supplemental information.

## Statistical analysis

Unpaired two-tailed Student's $t$-test and one- and two-way analysis of variance (ANOVA) were used to calculate $P$ values by GraphPad Prism 8 unless otherwise indicated in the figure legends. The Tukey method was used to adjust multiple comparisons. All data represent mean ± SEM or mean ± SD. $P < 0.05$ was considered significant. ns no significant difference. Kaplan–Meier curves were used to depict survival function from lifetime data for human patients using the log-rank test. Mice were randomly grouped before different treatments. In vitro studies, cells or conditions were assigned randomly to each experimental group.

# Data availability

The RNA-seq data produced in this study were deposited in the public database Gene Expression Omnibus (GEO) GSE250169 (https://www.ncbi.nlm.nih.gov/geo/query/acc.cgi?acc=GSE250169). The previously published datasets reanalyzed in this study were

available in GEO through the accession codes GSE190967 (https://www.ncbi.nlm.nih.gov/geo/query/acc.cgi?acc=GSE190967) and GSE148355 (https://www.ncbi.nlm.nih.gov/geo/query/acc.cgi?acc=GSE148355). TCGA LIHC gene expression and survival data were downloaded from https://www.cancer.gov/abouut-nci/organization/ccg/research/structural-genomics/tcga. All other data generated or analyzed in this study are available within the article and its supplementary information files. Source data are provided with this paper.

The source data of this paper are collected in the following database record: biostudies:S-SCDT-10_1038-S44318-024-00201-6.

## Peer review information

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

## Acknowledgements

This work was supported in part by grants from the National Natural Science Foundation of China (82341013, 82322050, 92357301, 82130087, and 82072656).

## Author contributions

**Xiaoyu Qian**: Data curation; Formal analysis; Investigation; Visualization; Methodology; Writing—original draft. **Jin Cai**: Data curation; Validation; Investigation; Writing—review and editing. **Yi Zhang**: Investigation; Methodology. **Shengqi Shen**: Software; Methodology. **Mingjie Wang**: Formal analysis. **Shengzhi Liu**: Validation. **Xiang Meng**: Investigation. **Junjiao Zhang**: Validation. **Zijian Ye**: Investigation. **Shiqiao Qiu**: Investigation. **Xiuying Zhong**: Validation; Investigation; Project administration; Writing—review and editing. **Ping Gao**: Funding acquisition; Project administration; Writing—review and editing.

Source data underlying figure panels in this paper may have individual authorship assigned. Where available, figure panel/source data authorship is listed in the following database record: biostudies:S-SCDT-10_1038-S44318-024-00201-6.

## Disclosure and competing interests statement

The authors declare no competing interests.

# Expanded View Figures

**A**

**B**

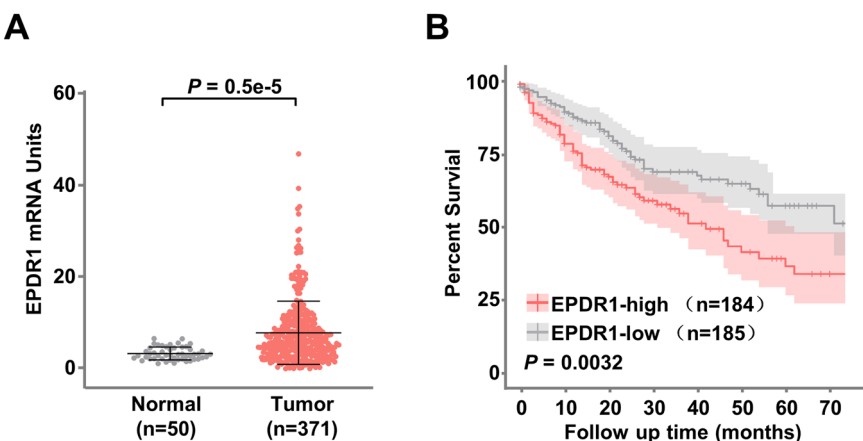

**Figure EV1.  Aberrant expression of EPDR1 in HCC is associated with immunosuppression.**

(**A**) The mRNA levels of EPDR1 were determined in adjacent noncancerous liver tissues (Normal) and liver cancer tissues (Tumor) from The Cancer Genome Atlas (TCGA) database, statistical analyses were performed by two-tailed unpaired Student's *t*-test. (**B**) Kaplan–Meier analysis of overall survival with log-rank tests for HCC patients with low versus high expression of EPDR1. Source data are available online for this figure.

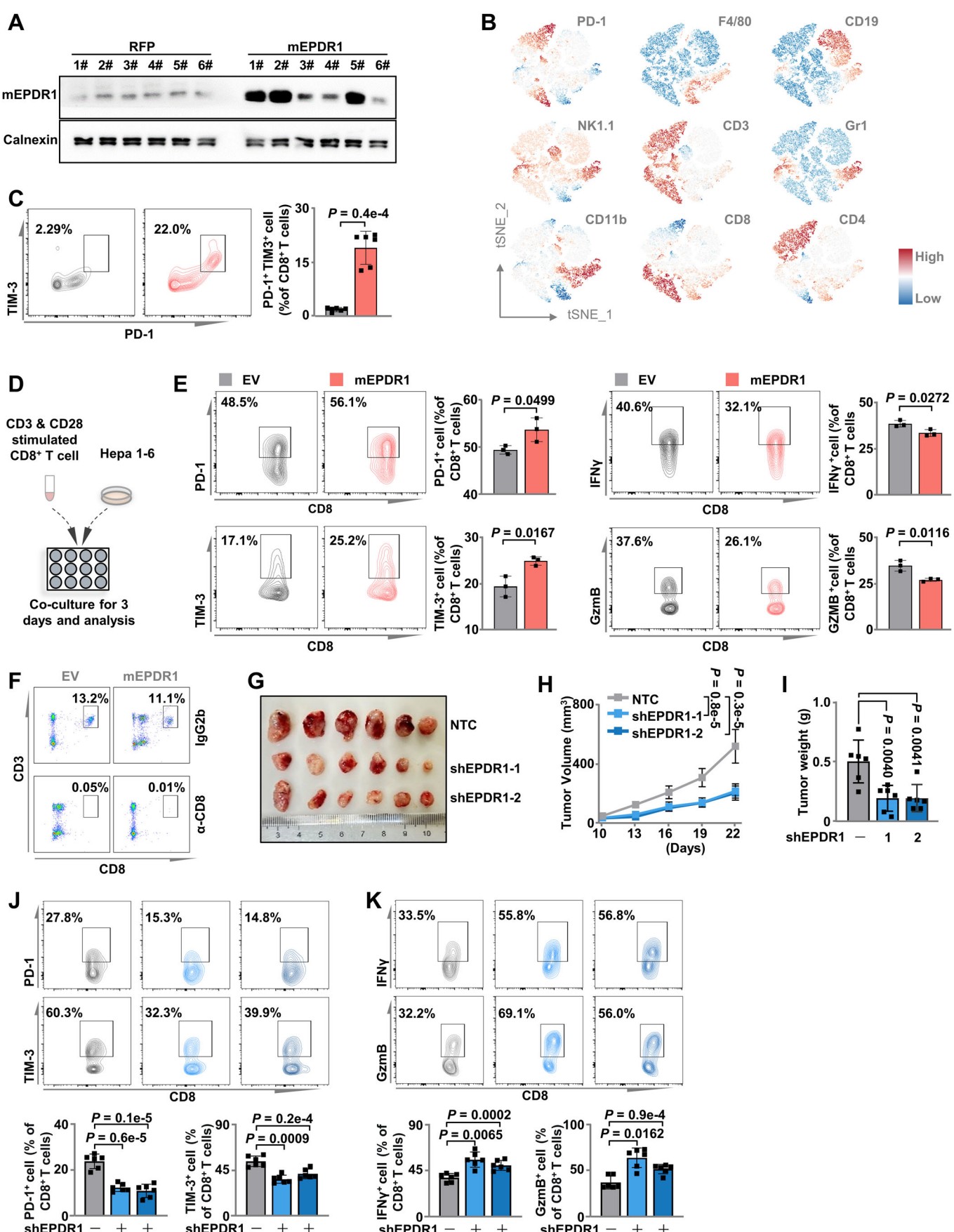

◀ **Figure EV2.   Tumor-intrinsic EPDR1 facilitates immune evasion by increasing antitumor CD8$^+$ T cells exhaustion.**

(**A**) Western blotting analysis of the protein level of Myc-tag mEPDR1 in liver samples from the YAP5SA-induced mouse model. Calnexin served as a loading control. (**B**) t-SNE plot showing the expression level marker genes for distinct subpopulations of lymphocytes. (**C**) Flow cytometric analysis of the ratio of immunosuppressive molecules (PD-1$^+$ TIM-3$^+$) cells in tumor CD8$^+$ T cells from the indicated group in (**A**). Data were presented as the mean ± SD. (**D**) Schema of coculture of mouse CD8$^+$ T cells with Hepa 1-6 cells expressing Flag-EV or Flag-mEPDR1. (**E**) Flow cytometry analysis of the ratio of immunosuppressive molecules (PD-1, TIM-3) and immune effector molecules (IFNγ, GzmB) positive cells in CD8$^+$ T cells after coculture with the indicated tumor cells. Data were presented as the mean ± SD of three independent experiments ($n = 3$). (**F**) Flowrate analysis of the amount of CD8$^+$ T cells in blood from the mice with the indicated manipulation. (**G–I**) Hepa 1–6 cells stably expressing NTC or shEPDR1 were injected subcutaneously into C57BL/6J mice ($n = 6$ male mice per group). Tumor size was measured starting at 10 days after inoculation. The figure depicts xenografts (**A**), growth curves (**B**), and tumor weights (**C**) determined at the end of the experiment (day 25). Data were presented as the mean ± SEM (**H**), and mean ± SD (**I**), respectively. (**J**) Flow cytometric analysis of the ratio of immunosuppressive molecules (PD-1, TIM-3) positive cells in tumor CD8$^+$ T cells from the indicated group in (**G**). Data were presented as the mean ± SD ($n = 6$). (**K**) Flow cytometric analysis of the ratio of immune effector molecules (IFNγ, Granzyme B) positive cells in tumor CD8$^+$ T cells from the indicated groups in (**G**). Data were presented as the mean ± SD ($n = 6$). Data information: Statistical significance was determined by two-tailed unpaired Student's *t*-test (**C, E**), two-way ANOVA (**H**), and one-way ANOVA (**I–K**). Source data are available online for this figure.

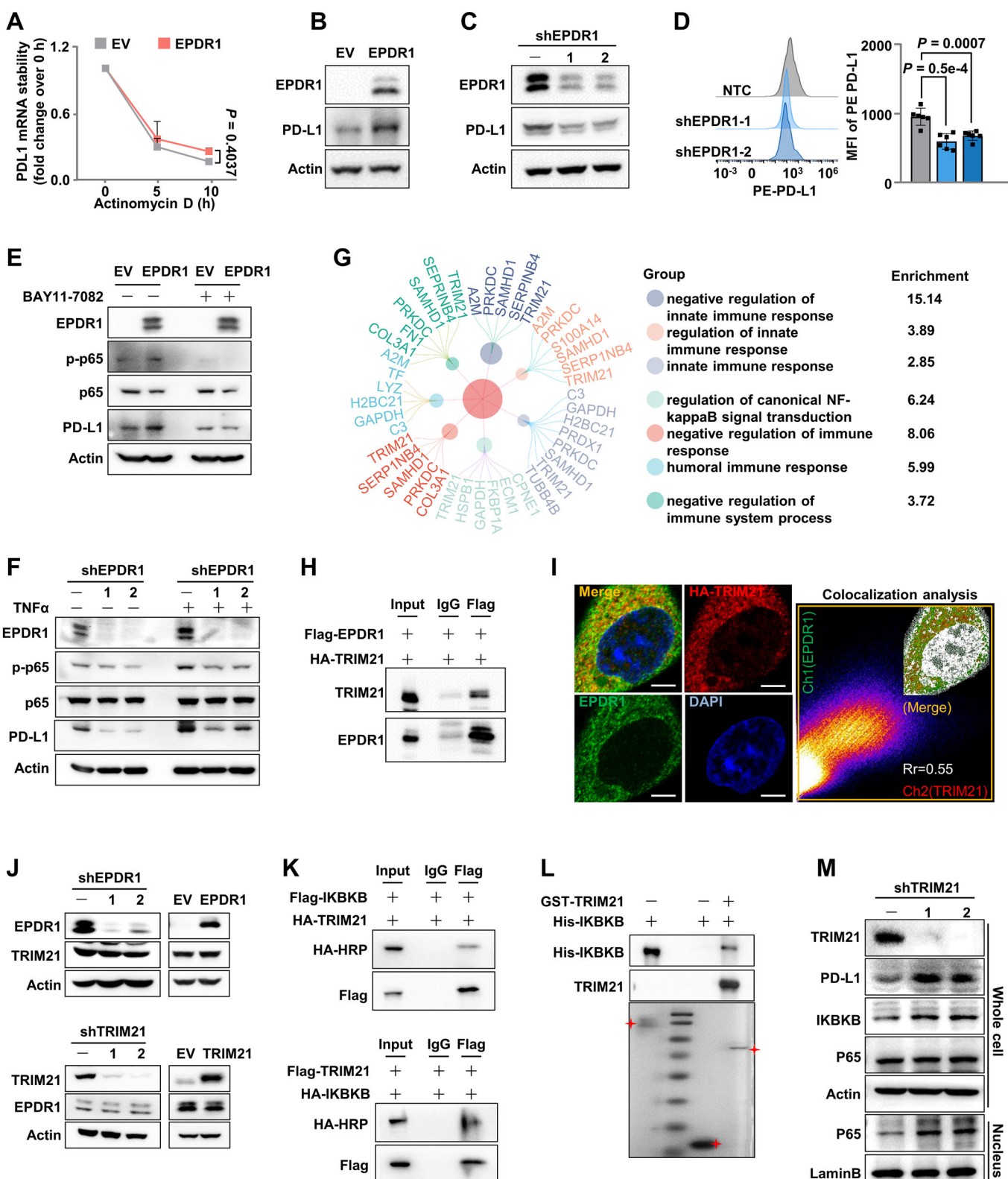

◀ **Figure EV3.  EPDR1 enhances the NF-κB pathway and elevates PD-L1 expression in cancer cells by interacting with TRIM21.**

(A) PD-L1 mRNA stability was determined in HepG2 cells expressing Flag-EV and Flag-EPDR1, cells treated with the transcription inhibitor actinomycin D (5 μM) or vehicle control for the indicated times ($n = 3$ biological replicates, data were presented as the mean ± SD, statistical significance was determined by two-way ANOVA). (B) Western blotting analysis of the protein levels of EPDR1 and PD-L1 in HepG2 cells expressing Flag-EV and Flag-EPDR1; β-actin was used as a loading control. (C) Western blotting analysis of the protein levels of EPDR1 and PD-L1 in HepG2 cells with EPDR1 knockdown; β-actin was used as a loading control. (D) Flow cytometry analysis of membrane binding PD-L1 on tumor cells separate from xenograft with mEPDR1 knockdown ($n = 6$ biological replicates, data are presented as the mean ± SD and statistical significance was determined by one-way ANOVA). (E) Western blotting analysis of the protein levels of PD-L1 and p65 in PLC cells expressing shNTC and shEPDR1. Cells were treated with the NF-κB agonist TNFα (100 nM) or vehicle control 6 h before sample collection, and β-actin was used as a loading control. (F) Western blotting analysis of the protein levels of PD-L1 and p65 in PLC cells expressing Flag-EV and Flag-EPDR1. Cells were treated with the NF-κB inhibitor BAY11-7082 (5 μM) or vehicle control 6 h before sample collection, and β-actin was used as a loading control. (G) Hierarchical network diagram showing the immune relate process that EPDR1 potential interactors involved in. (H) Co-IP assay showing the protein interaction between EPDR1 and TRIM21. HEK293T cells were transfected with Flag-EPDR1 and HA-TRIM21 plasmids. Cell lysates were immunoprecipitated with an anti-Flag antibody, followed by Western blotting analysis with antibodies against HA. (I) Representative images of immunofluorescence staining for endogenous EPDR1 and HA-TRIM21 in HepG2 cells. The nucleus was stained with DAPI; Scale bar, 20 μm. $R$ were calculated by Person's correlation analysis. (J) Western blotting analysis of the protein expression of EPDR1 or TRIM21 in HepG2 cells with the indicated genotypes. β-Actin was used as a loading control. (K) Co-IP assay showing the protein interaction between IKBKB and TRIM21. HEK293T cells were transfected with Flag-IKBKB plus HA-TRIM21 or Flag-TRIM21 plus HA-IKBKB plasmids. Cell lysates were immunoprecipitated with an anti-Flag antibody, followed by Western blotting analysis with antibodies against HA. (L) Pull-down assay showing the protein interaction between GST-TRIM21 and IKBKB-His. GST-tagged TRIM21 and 6× His-tagged IKBKB proteins were purified from *E. coli* and incubated in vitro, followed by Western blotting analysis with antibodies against TRIM21 or IKBKB. The red asterisks indicate the target bands. (M) Western blotting analysis of the protein levels of PD-L1, IKBKB, and p65 in whole cell lysate and nuclear fractions in HepG2 cells expressing shNTC and shTRIM21 with GAPDH and Lamin B, as the loading controls, respectively. Source data are available online for this figure.

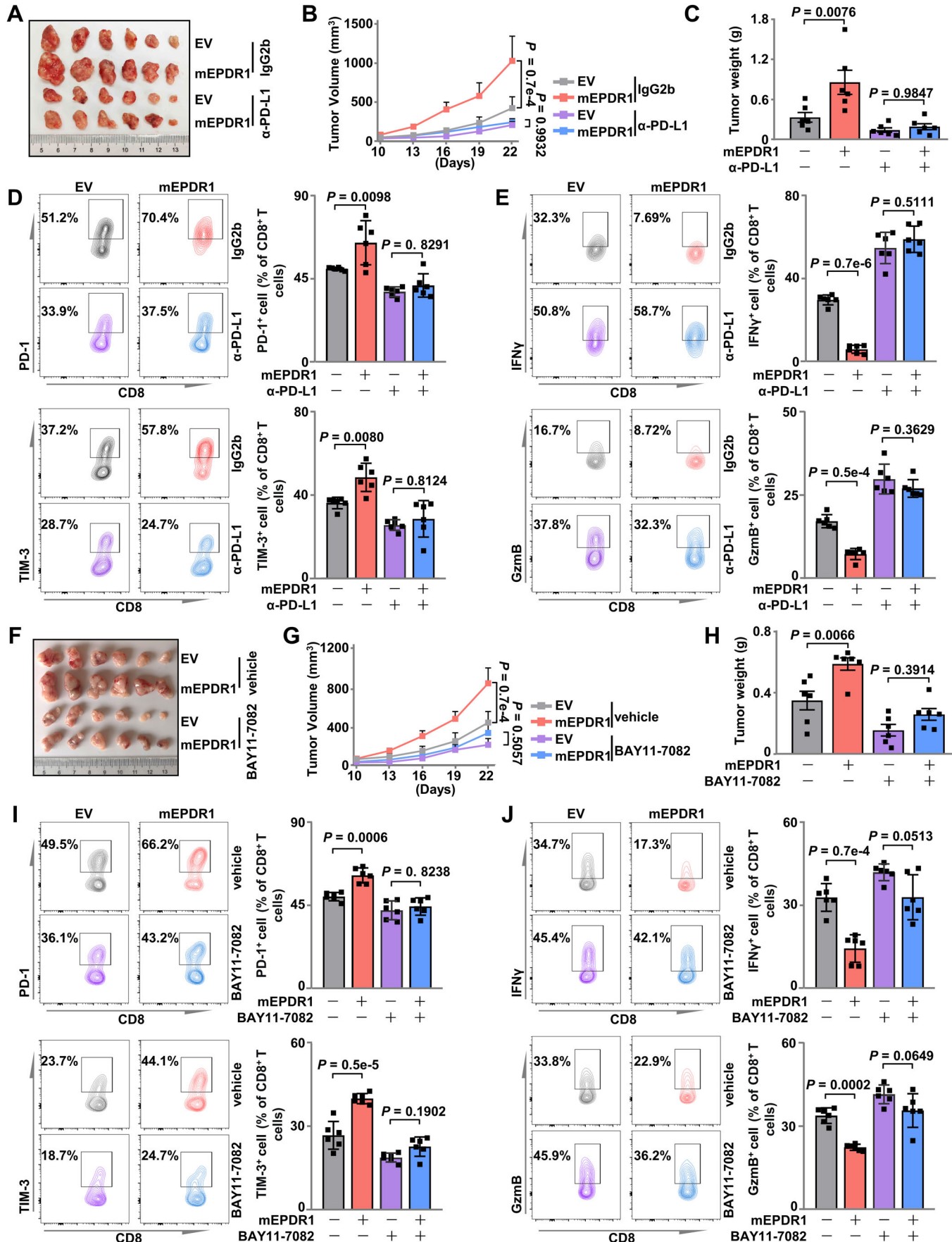

◀  **Figure EV4.   The EPDR1-TRIM21-PD-L1 axis promotes antitumor immune evasion in HCC.**

(**A–C**) Hepa 1-6 cells stably expressing Flag-EV and Flag-mEPDR1 were injected subcutaneously into C57BL/6J mice ($n = 6$ male mice per group), and α-PD-L1 (6 mg/kg) neutralizing antibody was injected intraperitoneally four times (twice a week starting at 10 days after inoculation) to block PD-L1 and IgG2b was used as control. Tumor size was measured starting at 10 days after inoculation. Photographs show xenografts (**A**), growth curves (**B**), and tumor weight (**C**) determined at the end of the experiment (day 25). Data were presented as the mean ± SEM (**B**) and mean ± SD (**C**), respectively. (**D**) Flow cytometry analysis of ratio of immunosuppressive molecules (PD-1, TIM-3) positive cells in tumor CD8$^+$ T cells from the indicated group in (**A**). Data were presented as the mean ± SD. (**E**) Flow cytometry analysis of the ratio of immune effector molecules (IFNγ, Granzyme B) positive cells in tumor CD8$^+$ T cells from the indicated groups in (**A**). Data were presented as the mean ± SD. (**F–H**) Hepa 1–6 cells stably expressing Flag-EV and Flag-mEPDR1 were injected subcutaneously into C57BL/6J mice ($n = 6$ male mice per group). BAY11-7082 was used to inhibit the NF-Kb pathway, and the vehicle was used as a control. Tumor size was measured starting at 10 days after inoculation. Photographs show xenografts (**F**), growth curves (**G**), and final tumor weight (**H**) determined at the end of the experiment (day 25). Data were presented as the mean ± SEM (**G**), and mean ± SD (**H**), respectively. (**I**) Flow cytometry analysis of the ratio of immunosuppressive molecules (PD-1, TIM-3) positive cells in tumor CD8$^+$ T cells from the indicated group in (**F**). Data were presented as the mean ± SD. (**J**) Flow cytometry analysis of the ratio of immune effector molecules (IFNγ, GzmB) positive cells in tumor CD8$^+$ T cells from the indicated group in (**F**). Data were presented as the mean ± SD. Data information: Statistical significance was determined by two-way ANOVA (**B**, **G**) and one-way ANOVA (**C–E**, **H–J**). Source data are available online for this figure.

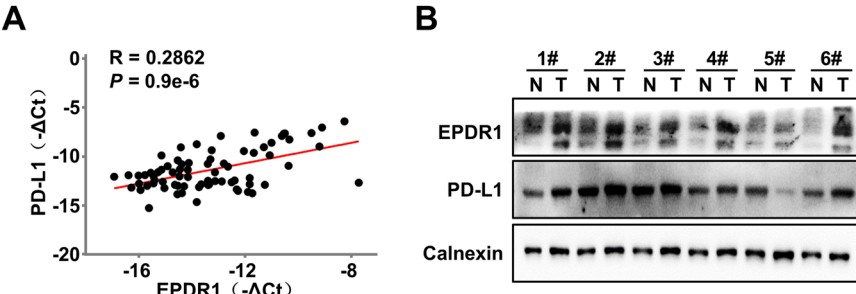

**Figure EV5. The level of EPDR1 expression is positively correlated with PD-L1 transcription in HCC tissues.**

(**A**) qRT-PCR analysis of the correlation of EPDR1 with PD-L1 in 20 pairs of clinically matched adjacent noncancerous liver tissues (Normal) and human liver cancer tissues (Tumor), *P* values and R were calculated by two-tailed Person's correlation analysis. (**B**) Western blotting analysis of the correlation of EPDR1 with PD-L1 in six pairs of matched adjacent noncancerous liver tissues (Normal) and mouse liver cancer tissues (Tumor). Source data are available online for this figure.

