## [Peer Review File · The EMBO Journal]

EPDR1 promotes PD-L1 expression and tumor immune evasion by inhibiting TRIM21-dependent ubiquitylation of I κ B kinase-beta

Xiaoyu Qian, Jin Cai, Yi Zhang, Shengqi Shen, Mingjie Wang, Shengzhi Liu, Xiang Meng, Junjiao Zhang, Zijian Ye, Shiqiao Qiu, Xiuying Zhong, and Ping Gao

Corresponding authors: Ping Gao (pgao2@ustc.edu.cn) , Xiuying Zhong (zxywawj@ustc.edu.cn)

Review Timeline:

Submission Date:	6th Dec 23
Editorial Decision:	20th Feb 24
Revision Received:	24th May 24
Editorial Decision:	21st Jun 24
Revision Received:	30th Jun 24
Accepted:	22nd Jul 24

Editor: Daniel Klimmeck

Transaction Report:

Dear Dr Ping Gao,

Thank you again for the submission of your manuscript (EMBOJ-2023-116324) to The EMBO Journal. As mentioned earlier, your study was assessed by three reviewers with expertise in tumor immunology and inflammatory signaling, whose comments are enclosed below.

As you will see from the experts' reports, the referees acknowledge the analysis and potential interest and value of your findings. However, they also express major concerns regarding clarity and molecular depth of the findings, which need to be addressed thoroughly to make them supportive of publication in the EMBO Journal. The reviewers also raise a number of issues related to the presentation of the findings, additional controls and improved methods annotation required, statistics applied and overall discussion of related literature, that would need to be conclusively addressed to achieve the level of robustness and clarity needed for The EMBO Journal.

Given the overall interest stated and broader angle of your findings, we are able to invite you to revise your manuscript experimentally to address the referees' comments. I need to stress though that we do require strong support from the referees on a revised version of the study in order to move on to publication of the work, and i.p. the concerns regarding a consolidated role of PD-L1 as functional EPDR1 target would need to be addressed in order to make this a compelling case to move forward with for the EMBO Journal.

In light of the extensive experimentation requested, I would appreciate if you could contact me during the next weeks for exchange e.g. a video call to discuss your perspective on the comments and potential plan for revisions.

Please feel free to contact me if you have any questions or need further input on the referee comments.

When submitting your revised manuscript, please carefully review the instructions below.

Please feel free to approach me any time should you have additional questions related to this.

Thank you for the opportunity to consider your work for publication.

I look forward to your revision.

Best regards,

Daniel Klimmeck

Daniel Klimmeck, PhD
Senior Editor
The EMBO Journal

Instruction for the preparation of your revised manuscript:

- 1) a .docx formatted version of the manuscript text (including legends for main figures, EV figures and tables). Please make sure that the changes are highlighted to be clearly visible.
- 2) individual production quality figure files as .eps, .tif, .jpg (one file per figure).
- 3) a .docx formatted letter INCLUDING the reviewers' reports and your detailed point-by-point response to their comments. As part of the EMBO Press transparent editorial process, the point-by-point response is part of the Review Process File (RPF), which will be published alongside your paper.

4) a complete author checklist, which you can download from our author guidelines ([https://wol-prod-cdn.literatumonline.com/pb-assets/embo-site/Author Checklist%20-%20EMBO%20J-1561436015657.xlsx](https://wol-prod-cdn.literatumonline.com/pb-assets/embo-site/Author%20Checklist%20-%20EMBO%20J-1561436015657.xlsx)). Please insert information in the checklist that is also reflected in the manuscript. The completed author checklist will also be part of the RPF.

6) It is mandatory to include a 'Data Availability' section after the Materials and Methods. Before submitting your revision, primary datasets produced in this study need to be deposited in an appropriate public database, and the accession numbers and database listed under 'Data Availability'. Please remember to provide a reviewer password if the datasets are not yet public (see <https://www.embopress.org/page/journal/14602075/authorguide#datadeposition>).

7) Our journal encourages inclusion of *data citations in the reference list* to directly cite datasets that were re-used and obtained from public databases. Data citations in the article text are distinct from normal bibliographical citations and should directly link to the database records from which the data can be accessed. In the main text, data citations are formatted as follows: "Data ref: Smith et al, 2001" or "Data ref: NCBI Sequence Read Archive PRJNA342805, 2017". In the Reference list, data citations must be labeled with "[DATASET]". A data reference must provide the database name, accession number/identifiers and a resolvable link to the landing page from which the data can be accessed at the end of the reference. Further instructions are available at .

8) At EMBO Press we ask authors to provide source data for the main and EV figures. Our source data coordinator will contact you to discuss which figure panels we would need source data for and will also provide you with helpful tips on how to upload and organize the files.

Numerical data can be provided as individual .xls or .csv files (including a tab describing the data). For 'blots' or microscopy, uncropped images should be submitted (using a zip archive or a single pdf per main figure if multiple images need to be supplied for one panel). Additional information on source data and instruction on how to label the files are available at .

9) We replaced Supplementary Information with Expanded View (EV) Figures and Tables that are collapsible/expandable online (see examples in <https://www.embopress.org/doi/10.15252/emboj.201695874>). A maximum of 5 EV Figures can be typeset. EV Figures should be cited as 'Figure EV1, Figure EV2' etc. in the text and their respective legends should be included in the main text after the legends of regular figures.

11) For data quantification: please specify the name of the statistical test used to generate error bars and P values, the number (n) of independent experiments (specify technical or biological replicates) underlying each data point and the test used to calculate p-values in each figure legend. The figure legends should contain a basic description of n, P and the test applied. Graphs must include a description of the bars and the error bars (s.d., s.e.m.).

Please remember: Digital image enhancement is acceptable practice, as long as it accurately represents the original data and conforms to community standards. If a figure has been subjected to significant electronic manipulation, this must be noted in the figure legend or in the 'Materials and Methods' section. The editors reserve the right to request original versions of figures and

the original images that were used to assemble the figure.

The revision must be submitted online within 90 days; please click on the link below to submit the revision online before 20th May 2024.

Referee #1:

In the manuscript titled "EPDR1 promotes PD-L1 expression and tumor immune evasion by inhibiting TRIM21-dependent ubiquitylation of IKBKB", the authors show that EPDR1 promotes tumor immune suppression through an increase of PD-L1 in the HCC tumor and an increase in T cell exhaustion. EPDR1 overexpression increases PD-L1 and its knockdown decreases PD-L1. EPDR1 interaction with TRIM21 prevents TRIM21 from degrading IKBKB which promotes IKBKB-NF- κ B dependent transcription of PD-L1. In the literature, it is known that TRIM21 can degrade PD-L1 and TRIM21 promotes NF- κ B signaling.

Major comments

1.The authors find that knockdown of EPDR1 decreases PD-L1 through RT-PCR technique in Fig 3A. In the result, several other important immune regulatory molecules are downregulated upon EPDR1 knockdown. Why do the authors select PD-L1 specifically as the major downstream target of EPDR1. The technique does not appear to be RT-PCR as stated by the authors but rather a heatmap of RNA-sequencing result. Was the PD-L1 reduction also confirmed by RT-PCR after the RNA-sequencing. The reduction of PD-L1 has been observed in western blot. Was the cell surface expression of membrane bound PD-L1 reduced after EPDR1 knockdown.

2.It is not clear how TRIM21 was identified to be a major interactor with EPDR1. Other proteins are noted as potential interactors as well, were those tested for binding with EPDR1. Why was TRIM21 selected.

3.The authors state that EPDR1 promotes immune suppression. However, the authors do not find differences in the tumor infiltrating T cell populations between EV and mEPDR1 (overexpressed EPDR1) after anti-PD-L1 antibody. Are there differences between mice injected with Hepa1-6-EPDR1 knockdown tumors and mice treated with anti-PD-L1 blockade.

Minor comments

1.The literature on TRIM21 and PD-L1 degradation, TRIM21 and NF- κ B is not cited.

2.In Fig S3K, the PD-L1 whole cell blot appears overexposed.

3.Line 338 states mouse CD8 T cells but it should be human.

Referee #2:

This study found that EPDR1 was upregulated in human HCC samples and EPDR1 mediated the expression of PDL1 in tumors, and they used HCC as a primary working model to analyze the mechanism by which EPDR1 regulate the exhaustion of CD8 T cell effector function. This work is interesting if EPDR1's function is really effective. I have several concerns for this work:

1.EPDR1 is intrinsically expressed in tumors, but EPDR1 is importantly to mediate the exhaustion of CD8+T cells. My questions, how EPDR1 engages in mediating the exhaustion of CD8+T cells? For example, how expression of EPDR1 in tumors may upregulate the expression of TIM-3 on CD8+T cells?

2.The percentage levels of intracellular IFN-gamma and granzyme B expression by CD8+ T cells are pretty high (e.g.78.4%). This is pretty surprising. Can they double check their cytokine/cytotoxic molecule assays are correct?

3.It appeared that IHC analyses of human HCC samples are nonspecific enough. The staining patterns of EPDR1, PD-L1 and P65 are almost the same.

4.They showed that EPDR1 promotes the activation of NF- κ B pathway. However, the NF- κ B activation may lead to a non-specific, global up-regulation of many inflammatory genes. How this could lead to deregulation of pro-inflammatory cytokines?

Referee #3:

In this manuscript Qian et al. tried to address the mechanism underlying immuno-suppressive effect of tumor intrinsic EPDR1-TRIM21-NF- κ B axis through promoting PD-L1 expression. The molecular interaction among EPDR1-TRIM21-NF- κ B activation and subsequent regulation of PD-L1 transcription were well demonstrated. Moreover, as previously reported, the authors showed that EPDR1 over-expression elicited tumor progressive activity in immune-competed mice but did not in immune compromised mice, which was supported by the induction of exhausted T cells in vivo. The authors also showed obvious

association of higher expression of EPDR1 with increased PD-L1 expression in tumor sites of HCC specimens. However, throughout the manuscript, data shown in this study did not sufficiently address how much PD-L1 expression contribute to EPDR1-NFkB axis-mediated immunosuppression on T cell activation. The possibility that cancer cell-derived immunosuppressive factors other than PD-L1 contributed to the growth advantage of EPDR1-high expressing cancer cells in vivo has not been evaluated. Therefore the descriptive data are still unsatisfactory to conclude the authors' claims. The authors should perform more comparative experiments to strengthen their claims. Specific comments are listed below.

Comment #1.

Activation of NF-kB, shown to be downstream of EPDR1 in this study, also regulates the expression of immunosuppressive or tumor-promoting factors such as IL-6 and IL-8 in addition to the PD-L1 expression. To clarify the significance of EPDR1-mediated immunosuppressive effects, expression profile of broad immunosuppressive and/or tumor-promoting factors in EPDR1-overexpressing tumor cells or in that tumor-bearing mice should be assessed.

Comment #2.

As shown in supple Fig. S4E-I, tumor retardation by the treatment with NF-kB inhibitor is quite reasonable, because NF-kB activity is responsible for the regulation of many types of immunosuppressive molecules in tumor-bearing hosts. It is critical to determine whether up-regulation of PD-L1 is dominantly and exclusively responsible for the growth advance of ERPM1-overexpressing tumor cells, or not. To address this issue, growth advantage of EPDR1-overexpressed and PD-L1-deficient Hepa1 cells should be assessed in vivo by comparing them with that of EPDR1-intact PD-L1 KO tumor cells. This analysis can also discriminate the effect of EPDR1-mediated regulation of PD-L1 expression from effects of EPDR1 on other immunosuppressive factors.

Comment #3.

To confirm the result of mouse study in Fig. 4A-E, it is critical to determine whether the exhausted phenotypes of human T cells that induced by EPDR1-overexpressing HCC are rescued by anti-PD-L1 blockade in the same experimental settings. This analysis may also prove how much EPDR1-NFkB axis-mediated PD-L1 expression contribute to immunosuppressive effect on T cell activation.

Comment #4.

PD-L1 expression is also detected not only on tumor cells but also on tumor-infiltrating immune cells such as macrophages and immature myeloid cells. In Fig. 5, was expression levels of PD-L1 expression analyzed only in cancerous but not in noncancerous tissues? The authors should evaluate the expression levels in each location of histological sections, as in Fig. 1. The results from noncancerous tissues may support the authors claim regarding the intrinsic effect of ERPM1 on PD-L1 expression.

Comment #5.

To prove the immunosuppressive effect of EPDR1 on human T cells in clinical setting, association of EPDR1 expression with infiltration of CD8 T cells and/or Treg into tumor sites should be evaluated, as the PD-L1 expression was assessed in Fig. 5.

Comment #6.

The results of Fig. 1A suggest the patients with higher expression of EPDR exhibit better prognosis. As did in supple. Fig. S1B, the authors should show the prognosis of EPDR1high and EPDR1low patients with or without benefit from ICB, to support the authors' claim that higher expression EPDR1 could be a predictive factor for better response to anti-PD-(L)1 therapy.

Comment #7.

Many types of cancer cells have been reported to express higher levels of EPDR1. Is EPDR1-mediated PD-L1 up-regulation the general immunosuppressive mechanism for cancer progression other than HCC? Or is that specific for HCC? Immunosuppressive effect of EPDR1-overexpressing or knocked down non-HCC cancer lines in vivo (Nude/ B6) can be evaluated in syngeneic rather than xenograft Hepa1-6 model. This issue is critical for considering broad application for many types of cancer.

Comment #8.

Any data demonstrating that ERPM1-mediated regulation of TRIM21 activity really contribute to the up-regulation of PD-L1 on tumor cells in vivo are missing. The comparative analysis of PD-L1 expression on the cell surface should be shown in both ERPM1-overexpressing (+/- Trim21 co-transfection) and ERPM1-knocked down tumor cells with flow cytometric analysis.

Comment #9

Regulation of TRIM21 level alone modulated IKBKB ubiquitination in vitro (Fig. 3 and FigS3) and seemed to ameliorate the exhausted phenotype of CD8 T cells in vivo (Fig. 4). Did overexpression of TRIM21 alone affect the PD-L1 expression on the surface, and other immunosuppressive molecules? This is related to the comment #1.

point-by-point response to the comments

Summary of Editor's comments:

Thank you again for the submission of your manuscript (EMBOJ-2023-116324) to The EMBO Journal. As mentioned earlier, your study was assessed by three referees with expertise in tumor immunology and inflammatory signaling, whose comments are enclosed below.

As you will see from the experts' reports, the referees acknowledge the analysis and potential interest and value of your findings. However, they also express major concerns regarding clarity and molecular depth of the findings, which need to be addressed thoroughly to make them supportive of publication in the EMBO Journal. The referees also raise a number of issues related to the presentation of the findings, additional controls and improved methods annotation required, statistics applied and overall discussion of related literature, that would need to be conclusively addressed to achieve the level of robustness and clarity needed for The EMBO Journal.

Given the overall interest stated and broader angle of your findings, we are able to invite you to revise your manuscript experimentally to address the referees' comments. I need to stress though that we do require strong support from the referees on a revised version of the study in order to move on to publication of the work, and i.p. the concerns regarding a consolidated role of PD-L1 as functional EPDR1 target would need to be addressed in order to make this a compelling case to move forward with for the EMBO Journal. As to the open outcome of the major revisional work required, I suggest keeping EMBO Reports in mind for this study as an alternative venue.

Response: We are very grateful for your kind decision to encourage us to proceed with this revision. We also thank the referees for their insightful comments and constructive suggestions. We believe that all the concerns are critical and relevant and will help us substantially strengthen the study.

Over the past three months, we have conducted additional experiments to address all the concerns and comments raised by our referees. We are now submitting a significantly improved manuscript along with our point-by-point response. In response to the referees' concerns, we have included all the relevant figures in the file, which we have labeled as **Figure R1** to **Figure R14**.

Point-by-point response to the comments of the Referees

Referee #1:

In the manuscript titled "EPDR1 promotes PD-L1 expression and tumor immune evasion by inhibiting TRIM21-dependent ubiquitylation of IKBKB", the authors show that EPDR1 promotes tumor immune suppression through an increase of PD-L1 in the HCC tumor and an increase in T cell exhaustion. EPDR1 overexpression increases PD-L1 and its knockdown decreases PD-L1. EPDR1 interaction with TRIM21 prevents TRIM21 from degrading IKBKB which promotes IKBKB-NF- κ B dependent transcription of PD-L1. In the literature, it is known that TRIM21 can degrade PD-L1 and TRIM21 promotes NF- κ B signaling.

Response: We are grateful for the referee's comments that well summarized the major findings and significance of our study. We also appreciate his/her constructive comments and suggestions for this manuscript.

Comments1-1:

The authors find that knockdown of EPDR1 decreases PD-L1 through RT-PCR technique in Fig 3A. In the result, several other important immune regulatory molecules are downregulated upon EPDR1 knockdown. Why do the authors select PD-L1 specifically as the major downstream target of EPDR1. The technique does not appear to be RT-PCR as stated by the authors but rather a heatmap of RNA-sequencing result. Was the PD-L1 reduction also confirmed by RT-PCR after the RNA-sequencing. The reduction of PD-L1 has been observed in western blot. Was the cell surface expression of membrane bound PD-L1 reduced after EPDR1 knockdown.

Response: We thank the referee for the important comments. In Figure 3A in the original manuscript, we detected the mRNA levels of immune checkpoint markers in HepG2 cells expressing shEPDR1s or NTC by qRT-PCR. The results indicated that EPDR1 knockdown significantly reduced the PD-L1 expression. Following the referee's suggestion, we have repeated this assay and presented the data in bar graph in Figure 3A in the revised manuscript. For your convenience, we have appended the revised figure here as **Figure R1A**. We further analyzed the cell surface expression of membrane-bound PD-L1 using flow-cytometric analysis. Consistent with the qRT-PCR result, EPDR1 knockdown in HepG2 cells significantly attenuated PD-L1 cell surface expression (**Figure R1B**), while EPDR1 overexpression remarkably increased its cell surface expression (**Figure R1C**). These results thus demonstrated that EPDR1 enhanced PD-L1 expression in HCC cells.

Figure R1. EPDR1 promotes PD-L1 expression in HCC cells.

A qRT-PCR analysis of the mRNA levels of a series of immune-related molecules in HepG2 cells with EPDR1 knockdown.

B Flow cytometric analysis of the membrane-bound PD-L1 in HepG2 cells with EPDR1 knockdown.

C Flow cytometric analysis of the membrane-bound PD-L1 in HepG2 cells expressing Flag-EPDR1 or Flag-EV

Data information: $n = 3$ independent experiments and the data are presented as the mean \pm SD in (A-C). Statistical significance was determined by two-way ANOVA(A), one-way ANOVA (B) and two-tailed unpaired Student's t -test unpaired (C).

Comments1-2:

It is not clear how TRIM21 was identified to be a major interactor with EPDR1. Other proteins are noted as potential interactors as well, were those tested for binding with EPDR1. Why was TRIM21 selected.

Response: We appreciate the referee's concern. To explore the possible mechanism by which EPDR1 promotes p65 nuclear translocation and PD-L1 transcriptional upregulation, we conducted immunoprecipitation (IP) assays with antibody targeting Flag-tagged EPDR1 in HepG2 cells followed by MS identification of candidate interaction partners. In particular, TRIM21 that was the only one candidate reportedly involved in regulating the NF- κ B pathway among the 6 detected highly probable interactor of EPDR1 (**Figure EV3G** in the revised manuscript) (Gullà *et al*, 2018; Huang *et al*, 2024; Wada *et al*, 2009). More importantly, TRIM21 overexpression resulted in decreased IKBKB protein levels and reduced p65 nuclear translocation in HepG2 cells, and this effect could be reversed by

EPDR1 overexpression (**Figure 3M** in the original manuscript). These data indicated that TRIM21 was indeed required for EPDR1-mediated NF- κ B/PD-L1 activation.

Comments1-3:

The authors state that EPDR1 promotes immune suppression. However, the authors do not find differences in the tumor infiltrating T cell populations between EV and mEPDR1 (overexpressed EPDR1) after anti-PD-L1 antibody. Are there differences between mice injected with Hepa 1-6-EPDR1 knockdown tumors and mice treated with anti-PD-L1 blockade.

Response: We thank the referee for the critical comments. To investigate whether PD-L1 participated in EPDR1-mediated tumor immune evasion, we employed a xenograft model mice generated by subcutaneous inoculation with murine liver cancer cell Hepa 1-6 overexpressing EPDR1 or EV and quantified exhaustion and activity of tumor-infiltrating CD8⁺ T cells in tumor-bearing mice treated with or without anti-PD-L1 antibody. Flow cytometric analysis revealed that the proportion of exhausted CD8⁺ T cells (*i.e.*, PD-1⁺ or TIM-3⁺) was significantly increased and CD8⁺ T cell activity (reflected by the proportion of IFN γ ⁺ or Granzyme B⁺ (GzmB⁺) cells) was obviously decreased in mice inoculating with Hepa 1-6 cells overexpressing EPDR1 compared to that in mice inoculating with EV cells, and this effect could be abolished by PD-L1 monoclonal neutralizing antibody treatment (**Figure 4D and E** in the original manuscript), suggesting that PD-L1 contributed to EPDR1-mediated immune evasion.

Following the referee's suggestion, we used the PD-L1 monoclonal neutralizing antibody to treat immune-competent mice subcutaneously inoculated with EPDR1 knockdown or NTC-expressing Hepa 1-6 cells and examined the tumor infiltrating T cell populations in tumor-bearing mice. EPDR1 knockdown and anti-PD-L1 treatment each significantly attenuated tumor growth (**Figure R2A-C**). Flow cytometric analysis showed that EPDR1 inhibition and anti-PD-L1 each resulted in significant suppression of exhaustion and increased antitumor activity in tumor-infiltrating CD8⁺ T cells (**Figure R2D and E**). The infiltrated CD8⁺ T cells exhibited slightly but not significantly, lower PD-1 and TIM-3 expression and higher IFN γ and GzmB expression in mice inoculated with EPDR1 knockdown Hepa 1-6 cells compared with those treated with anti-PD-L1. The probable reason is that PD-L1 is not only expressed in cancer cells but also in tumor associated stromal cells. These data indicated that loss of EPDR1 in cancer cells enhanced the antitumor activity of tumor-infiltrating CD8⁺ T cells.

Figure R2. EPDR1 enhances PD-L1-mediated immune evasion of HCC cells.

A-C Hepa 1-6 cells stably expressing NTC or shEPDR1 were injected subcutaneously into C57BL/6J mice (n = 6 male mice per group), and α -PD-L1 (6 mg/kg) neutralizing antibody was injected intraperitoneally 4 times (twice a week starting at 10 days after inoculation) to block PD-L1 and IgG2b was used as control. Tumor size was measured starting at 10 days after inoculation. Figures depict xenografts (A), growth curves (B) and tumor weight (C) determined at the end of the experiment (day 25). Data are presented as the mean \pm SEM.

D Flow cytometric analysis of ratio of immunosuppressive molecules (PD-1, TIM-3) positive cells in tumor CD8⁺ T cells from the indicated group in (A). Data are presented as the mean \pm SD.

E Flow cytometric analysis of ratio of immune effector molecules (IFN γ , GzmB) positive cells in tumor CD8⁺ T cells from the indicated group in (A). Data are presented as the mean \pm SD.

Data information: Statistical significance was determined two-way ANOVA (B) and one-way ANOVA (C-E).

To further demonstrate that PD-L1 is functional EPDR1 target, we inoculated EPDR1 or EV overexpressing Hepa 1-6 cells with PD-L1 knockdown into mice and detected tumor-infiltrating CD8⁺ T cell exhaustion and activity markers. We found that EPDR1 overexpression facilitated the *in vivo* tumor growth of Hepa 1-6 xenografts, which was largely abolished by PD-L1 knockdown (Figure R3A-C). Further flow cytometric analysis revealed that EPDR1 overexpression in cancer cells resulted in the increased exhaustion and decreased activity of tumor-infiltrating T cells in mice, and this effect was abolished by PD-L1 knockdown (Figure R3D and E). Collectively, these data indicated that EPDR1-

PD-L1 axis in cancer cells promotes tumor immune evasion by increasing exhaustion of tumor-infiltrating CD8⁺ T cells.

Figure R3. EPDR1-PD-L1 axis in cancer cells promotes tumor immune evasion by increasing exhaustion of tumor-infiltrating CD8⁺ T cells.

A-C Hepa 1-6 cells stably expressing Flag-EV and Flag-mEPDR1 were infected with shNTC or shPD-L1, and these cells were injected subcutaneously into C57BL/6J mice (n = 6 male mice per group). Tumor size was measured starting at 10 days after inoculation. Figures depict xenografts (A), growth curves (B) and tumor weight (C) determined at the end of the experiment (day 25). Data are presented as the mean \pm SEM.

D Flow cytometric analysis of ratio of immunosuppressive molecules (PD-1, TIM-3) positive cells in tumor CD8⁺ T cells from the indicated group in (A). Data are presented as the mean \pm SD.

E Flow cytometric analysis of ratio of immune effector molecules (IFN γ , Granzyme B) positive cells in tumor CD8⁺ T cells from the indicated groups in (A). Data are presented as the mean \pm SD.

Data information: Statistical significance was determined two-way ANOVA (B) and one-way ANOVA (C-E).

Comments1-4:

The literature on TRIM21 and PD-L1 degradation, TRIM21 and NF- κ B is not cited.

Response: We thank the referee for the constructive advice and have now discussed these important points in the Results and Discussion sections and have cited the literature in the revised manuscript.

Comments1-5:

In Fig S3K, the PD-L1 whole cell blot appears overexposed.

Response: We appreciate the referee's critical comments. We replaced a shorter expose blot of PD-L1 in the revised figure (**Figure R4A**, also see as **Figure EV3M** in the revised manuscript). We also append here another independent western blot to verify that TRIM21 knockdown resulted in increased expression of IKBKB and PD-L1 and reduced p65 nuclear translocation (**Figure R4B and C**).

Figure R4. TRIM21 knockdown results in increased expression of IKBKB and PD-L1 and reduced p65 nuclear translocation.

A Western blotting analysis of the protein levels of PD-L1, IKBKB and p65 in whole cell lysate and nuclear fractions in HepG2 cells with TRIM21 knockdown, ACTIN and Lamin B were used as the loading controls, respectively.

B, C Twice repeat western blotting results of experiment A

Comments1-6:

Line 338 states mouse CD8 T cells but it should be human.

Response: Thank you for pointing this out. We have corrected the description in the revised manuscript.

Referee #2:

This study found that EPDR1 was upregulated in human HCC samples and EPDR1 mediated the expression of PDL1 in tumors, and they used HCC as a primary working model to analyze the

mechanism by which EPDR1 regulate the exhaustion of CD8 T cell effector function. This work is interesting if EPDR1's function is really effective.

Response: We thank the referee for your encouraging and constructive comments.

Comments2-1:

EPDR1 is intrinsically expressed in tumors, but EPDR1 is importantly to mediate the exhaustion of CD8+T cells. My questions, how EPDR1 engages in mediating the exhaustion of CD8+T cells? For example, how expression of EPDR1 in tumors may upregulate the expression of TIM-3 on CD8+T cells?

Response: This is a good point. Actually, we did not investigate whether tumor intrinsic EPDR1 could directly regulate the expression of TIM-3 in CD8⁺ T cells. In current study, by employing HCC mouse model and in vitro co-culture assay, we found that aberrant EPDR1 expression in cancer cells increased exhaustion levels and decreased antitumor activity of tumor-infiltrating CD8⁺ T cells. The expression of exhaustion markers (i.e., PD-1⁺ or TIM-3⁺) on tumor-infiltrating CD8⁺ T cells was analyzed to indicate the proportion of exhausted CD8⁺ T cells (Kersten *et al*, 2022; Tang *et al*, 2023; Zhang *et al*, 2024) Mechanistic study demonstrated that EPDR1 increases tumor-infiltrating CD8⁺ T cell exhaustion by activating PD-L1 in cancer cells. The immune outcomes from cancer cell surface PD-L1 engaging CD8⁺ T cell surface PD-1 are well described and are a basis for ICB antibodies (Ribas & Wolchok, 2018). We have performed additional experiments to demonstrate that PD-L1 is functional EPDR1 target in cancer cells. In xenograft model mice generated by subcutaneous inoculation with EPDR1 or EV overexpressing Hepa 1-6 cells with PD-L1 knockdown, we observed that EPDR1 overexpression facilitated the *in vivo* tumor growth of Hepa 1-6 xenografts, which was largely abolished by PD-L1 knockdown (**Figure R3A-C**, the Referee #1 also raised a correlated question. For your convenience, we append the figures here once again). Further flow cytometric analysis revealed that EPDR1 overexpression in cancer cells resulted in the increased exhaustion and decreased activity of tumor-infiltrating T cells in mice, and this effect was abolished by PD-L1 knockdown (**Figure R3D and E**). Collectively, these data indicated that aberrant EPDR1 expression in cancer cells promotes PD-L1-mediated tumor immune evasion by increasing exhaustion of tumor-infiltrating CD8⁺ T cells.

Figure R3. EPDR1-PD-L1 axis in cancer cells promotes tumor immune evasion by increasing exhaustion of tumor-infiltrating CD8⁺ T cells.

A-C Hepa 1-6 cells stably expressing Flag-EV and Flag-mEPDR1 were infected with shNTC or shPD-L1, and these cells were injected subcutaneously into C57BL/6J mice (n = 6 male mice per group). Tumor size was measured starting at 10 days after inoculation. Figures depict xenografts (A), growth curves (B) and tumor weight (C) determined at the end of the experiment (day 25). Data are presented as the mean \pm SEM.

D Flow cytometric analysis of ratio of immunosuppressive molecules (PD-1, TIM-3) positive cells in tumor CD8⁺ T cells from the indicated group in (A). Data are presented as the mean \pm SD.

E Flow cytometric analysis of ratio of immune effector molecules (IFN γ , Granzyme B) positive cells in tumor CD8⁺ T cells from the indicated groups in (A). Data are presented as the mean \pm SD.

Data information: Statistical significance was determined two-way ANOVA (B) and one-way ANOVA (C-E).

Comments2-2:

The percentage levels of intracellular IFN-gamma and granzyme B expression by CD8⁺ T cells are pretty high (e.g.78.4%). This is pretty surprising. Can they double check their cytokine/cytotoxic molecule assays are correct?

Response: We appreciate the referee for the critical comments. The flow cytometric analysis was performed as previously described (Ma *et al.*, 2019; Tang *et al.*, 2023; Zhang *et al.*, 2024). Following the referee's suggestion, we repeated this experiment. By employing a xenograft model mice generated by subcutaneous inoculation with Hepa 1-6 cells with differential expression levels of EPDR1 and TRIM21, we observed that EPDR1 overexpression facilitated the *in vivo* tumor growth of Hepa 1-6 xenografts, which was largely abolished by TRIM21 overexpression (**Figure R5A-C**). We also found that EPDR1 overexpression increased, while TRIM21 overexpression decreased, the surface expression of PD-L1 in tumor cells (**Figure R5D**). Further flow cytometric analysis showed that TRIM21 expression could reverse the increased exhaustion and decreased activity of tumor-infiltrating T cells in mice inoculated with Hepa 1-6 cells overexpressing EPDR1 (**Figure R5E and F**). These data indicated that TIM21 is involved in EPDR1-mediated tumor immune evasion.

Figure R5. TIM21 is involved in EPDR1-mediated tumor immune evasion.

A-C Hepa 1-6 cells stably expressing Flag-EV, Flag-mTRIM21, Flag-mEPDR1 or Flag-mTRIM21 plus Flag-mEPDR1 both were injected subcutaneously into C57BL/6J mice (n = 6 male mice per group). Tumor size was measured

starting at 10 days after inoculation. Figures depict xenografts (A), growth curves (B) and tumor weight (C) determined at the end of the experiment (day 25). Data are presented as the mean \pm SEM.

- D** Flow cytometric analysis of membrane-bound PD-L1 in the indicated group in (A). Data are presented as the mean \pm SD.
- E** Flow cytometric analysis of ratio of immunosuppressive molecules (PD-1, TIM-3) positive cells in tumor CD8⁺ T cells from the indicated group in (A). Data are presented as the mean \pm SD.
- F** Flow cytometric analysis of ratio of immune effector molecules (IFN γ , GzmB) positive cells in tumor CD8⁺ T cells from the indicated group in (A). Data are presented as the mean \pm SD.

Data information: Statistical significance was determined two-way ANOVA (B) and one-way ANOVA (C-F).

Comments2-3:

It appeared that IHC analyses of human HCC samples are nonspecific enough. The staining patterns of EPDR1, PD-L1 and p65 are almost the same.

Response: We appreciate the referee's valuable advice. The commercialized antibodies targeting EPDR1, PD-L1 and p65 used in this study are available for IHC. We also verified the specificity of the antibodies by western blot using HepG2 cells expressing shRNAs targeting EPDR1, PD-L1 or p65 (**Figure R6A**). In addition, we have performed additional IHC staining using serial sections of the same HCC tissues to evaluate the correlation between EPDR1 expression level and p65 nuclear location and PD-L1 expression in clinical HCC tissue (**Figure R6B-D**). IHC analysis of 50 clinical HCC tumor samples showed that EPDR1 protein levels were positively correlated with PD-L1 expression levels and nuclear p65 levels.

Figure R6. EPDR1 protein levels were positively correlated with PD-L1 expression levels and nuclear p65 levels in clinical HCC tumor samples.

A Western blot analysis of EPDR1, PD-L1 and p65 expression in HepG2 cells with the indicated genotypes. β -actin was used as a loading control.

B Representative immunohistochemistry images of EPDR1, PD-L1 and p65 staining in HCC specimens; scale bars, 50 μ m.

C Correlation analysis of EPDR1 and PD-L1 positive signal in HCC specimens, P -values and R were calculated by two-tailed Person's correlation analysis.

D Correlation analysis of EPDR1 and nucleus p65 positive signal in HCC specimens, P -values and R were calculated by two-tailed Person's correlation analysis.

Comments2-4:

They showed that EPDR1 promotes the activation of NF- κ B pathway. However, the NF- κ B activation may lead to a non-specific, global up-regulation of many inflammatory genes. How this could lead to deregulation of pro-inflammatory cytokines?

Response: Thank you for the important question. Using a series of knockdown and overexpression lines of HCC cells and HCC mouse models, we found that EPDR1 increases tumor-infiltrating CD8⁺ T cell exhaustion by activating NF-κB/PD-L1 axis in cancer cells. The expression of cytokines IFN γ and GzmB in tumor-infiltrating CD8⁺ T cells was analyzed to indicate their activity (Kersten *et al*, 2022; Tang *et al*, 2023; Zhang *et al*, 2024). Truly as our referee has pointed out, NF-κB has been known for their central role in coordinating immune and inflammation, and is not only expressed in immune cells but also in cancer cells. However, EPDR1 that we identified as an important regulator of NF-κB, was expressed at substantially higher levels in tumor cells compared to that in immune cells including T cells (**Figure R9C**). In this regard, our data suggest that EPDR1 may be a potential target for combinational immunotherapies.

Referee #3:

In this manuscript Qian et al. tried to address the mechanism underlying immuno-suppressive effect of tumor intrinsic EPDR1-TRIM21-NF-κB axis through promoting PD-L1 expression. The molecular interaction among EPDR1-TRIM21-NF-κB activation and subsequent regulation of PD-L1 transcription were well demonstrated. Moreover, as previously reported, the authors showed that EPDR1 over-expression elicited tumor progressive activity in immune-competed mice but did not in immune compromised mice, which was supported by the induction of exhausted T cells in vivo. The authors also showed obvious association of higher expression of EPDR1 with increased PD-L1 expression in tumor sites of HCC specimens. However, throughout the manuscript, data shown in this study did not sufficiently address how much PD-L1 expression contribute to EPDR1-NFκB axis-mediated immunosuppression on T cell activation. The possibility that cancer cell-derived immunosuppressive factors other than PD-L1 contributed to the growth advantage of EPDR1-high expressing cancer cells in vivo has not been evaluated. Therefore, the descriptive data are still unsatisfactory to conclude the authors' claims.

Response: We thank the referee for the positive opinion and encouraging comments that have well summarized this study. We also appreciate the referee for the constructive advice, which certainly help us strengthen our study. As these important concerns are further detailed in his/her major comments, we addressed them one by one below.

Comments3-1:

Activation of NF-κB, shown to be downstream of EPDR1 in this study, also regulates the expression of immunosuppressive or tumor-promoting factors such as IL-6 and IL-8 in addition to the PD-L1 expression. To clarify the significance of EPDR1-mediated immunosuppressive effects, expression

profile of broad immunosuppressive and/or tumor-promoting factors in EPDR1-overexpressing tumor cells or in that tumor-bearing mice should be assessed.

Response: We appreciate the referee for the critical comments. Following the referee's suggestion, we investigated the expression of NF- κ B downstream immunosuppressive or tumor-promoting factors in EPDR1-overexpressing HepG2 cells. qRT-PCR analysis showed that EPDR1 overexpression significantly increased the mRNA levels of PD-L1, IL-6 and IL-1 β (**Figure R7**). The PD-L1/PD-1 axis is a critical determinant of immune homeostasis and a basis for ICB antibodies. To demonstrate that PD-L1 contributes to EPDR1-mediated immune evasion, we have performed additional *in vivo* experiments. In xenograft model mice generated by subcutaneous inoculation with EPDR1 or EV overexpressing Hepa 1-6 cells with PD-L1 knockdown, we observed that EPDR1 overexpression facilitated the *in vivo* tumor growth of Hepa 1-6 xenografts, which was largely abolished by PD-L1 knockdown (**Figure R3A-C**, the Referee #1 also raised a correlated question. For your convenience, we append the figures here once again). Further flow cytometric analysis revealed that EPDR1 overexpression in cancer cells resulted in the increased exhaustion and decreased activity of tumor-infiltrating T cells in mice, and this effect was abolished by PD-L1 knockdown (**Figure R3D and E**). Collectively, these data indicated that aberrant EPDR1 expression in cancer cells promotes PD-L1-mediated tumor immune evasion by increasing exhaustion of tumor-infiltrating CD8⁺ T cells. Considering the critical contribution of IL-6 and IL-1 β in inflammation responses and tumor progression (Han *et al*, 2023; Johnson *et al*, 2018), the potential roles of IL-6 and IL-1 β in EPDR1-mediated tumor immune suppression warrant further independent study.

A

Figure R7. EPDR1 promoting NF- κ B signal and PD-L1 expression.

qRT-PCR analysis of mRNA level of a series NF- κ B target genes in HepG2 cells expressing EPDR1 or empty vector (n = 3, mean \pm SD, Statistical significance was determined by two-way ANOVA).

Figure R3. EPDR1-PD-L1 axis in cancer cells promotes tumor immune evasion by increasing exhaustion of tumor-infiltrating CD8⁺ T cells.

A-C Hepa 1-6 cells stably expressing Flag-EV and Flag-mEPDR1 were infected with shNTC or shPD-L1, and these cells were injected subcutaneously into C57BL/6J mice (n = 6 male mice per group), Tumor size was measured starting at 10 days after inoculation. Figures depict xenografts (A), growth curves (B) and tumor weight (C) determined at the end of the experiment (day 25). Data are presented as the mean \pm SEM.

D Flow cytometric analysis of ratio of immunosuppressive molecules (PD-1, TIM-3) positive cells in tumor CD8⁺ T cells from the indicated group in (A). Data are presented as the mean \pm SD.

E Flow cytometric analysis of ratio of immune effector molecules (IFN γ , Granzyme B) positive cells in tumor CD8⁺ T cells from the indicated groups in (A). Data are presented as the mean \pm SD.

Data information: Statistical significance was determined two-way ANOVA (B) and one-way ANOVA (C-E).

Comments3-2:

As shown in supple Fig. S4E-I, tumor retardation by the treatment with NF- κ B inhibitor is quite reasonable, because NF- κ B activity is responsible for the regulation of many types of immunosuppressive molecules in tumor-bearing hosts. It is critical to determine whether up-regulation of PD-L1 is dominantly and exclusively responsible for the growth advance of ERPM1-overexpressing

tumor cells, or not. To address this issue, growth advantage of EPDR1-overexpressed and PD-L1-deficient Hepa 1-6 cells should be assessed *in vivo* by comparing them with that of EPDR1-intact PD-L1 KO tumor cells. This analysis can also discriminate the effect of EPDR1-mediated regulation of PD-L1 expression from effects of EPDR1 on other immunosuppressive factors.

Response: We thank the referee for the constructive advice. Following our referee's suggestion, we inoculated EPDR1 or EV overexpressing Hepa 1-6 cells with PD-L1 knockdown into mice and detected tumor-infiltrating CD8⁺ T cell exhaustion and activity markers. We found that EPDR1 overexpression facilitated the *in vivo* tumor growth of Hepa 1-6 xenografts, which was largely abolished by PD-L1 knockdown (**Figure R3A-C** the Referee #1 also raised a correlated question. For your convenience, we append the figures here once again). Further flow cytometric analysis revealed that EPDR1 overexpression in cancer cells resulted in the increased exhaustion and decreased activity of tumor-infiltrating T cells in mice, and this effect was abolished by PD-L1 knockdown (**Figure R3D and E**). Similar results were observed in anti-PD-L1 treated xenograft model mice generated by subcutaneous inoculation with Hepa 1-6 cells overexpressing EPDR1 or EV control (**Figure EV4** in the revised manuscript). Collectively, these data indicated that EPDR1-PD-L1 axis in cancer cells promotes tumor immune evasion by increasing exhaustion of tumor-infiltrating CD8⁺ T cells.

Figure R3. EPDR1-PD-L1 axis in cancer cells promotes tumor immune evasion by increasing exhaustion of tumor-infiltrating CD8⁺ T cells.

A-C Hepa 1-6 cells stably expressing Flag-EV and Flag-mEPDR1 were infected with shNTC or shPD-L1, and these cells were injected subcutaneously into C57BL/6J mice (n = 6 male mice per group), Tumor size was measured starting at 10 days after inoculation. Figures depict xenografts (A), growth curves (B) and tumor weight (C) determined at the end of the experiment (day 25). Data are presented as the mean \pm SEM.

D Flow cytometric analysis of ratio of immunosuppressive molecules (PD-1, TIM-3) positive cells in tumor CD8⁺ T cells from the indicated group in (A). Data are presented as the mean \pm SD.

E Flow cytometric analysis of ratio of immune effector molecules (IFN γ , Granzyme B) positive cells in tumor CD8⁺ T cells from the indicated groups in (A). Data are presented as the mean \pm SD.

Data information: Statistical significance was determined two-way ANOVA (B) and one-way ANOVA (C-E).

Comments3-3:

To confirm the result of mouse study in Fig. 4A-E, it is critical to determine whether the exhausted phenotypes of human T cells that induced by EPDR1-overexpressing HCC are rescued by anti-PD-L1 blockade in the same experimental settings. This analysis may also prove how much EPDR1-NF κ B axis-mediated PD-L1 expression contribute to immunosuppressive effect on T cell activation.

Response: We appreciate the referee for the critical comments. Following the referee's suggestion, we performed additional co-culture experiment to investigate whether PD-L1 is involved in EPDR1-mediated tumor immune escape. Human CD8⁺ T cells that are isolated from peripheral blood mononuclear cells were pre-activated with anti-CD3 and anti-CD28 antibodies and co-cultured with HepG2 cells overexpressing EPDR1 or EV for 5 days. Anti-PD-L1 neutralizing antibody was introduced into the co-culture system. Consistent with our results in HCC mouse model, CD8⁺ T cells co-cultured with EPDR1-overexpressing HepG2 cells had higher levels of exhaustion (*i.e.*, PD-1⁺ or TIM-3⁺) and decreased antitumor activity (reflected by the proportion of IFN γ ⁺ or GzmB⁺ cells), which could be abolished by anti-PD-L1 blockade (**Figure R8A and B**), suggesting that EPDR1-mediated PD-L1 expression contributed to its immunosuppressive effect on CD8⁺ T cell activity.

Figure R8. PD-L1 contributes to EPDR1-mediated immunosuppressive effect on CD8⁺ T cell activity.

A Flow cytometric analysis of the ratio of immunosuppressive molecules (PD-1, TIM-3) positive cells in CD8⁺ T cells after coculture with the indicated tumor cells. n = 3 independent experiments.

B Flow cytometric analysis of the ratio of immune effector molecules (IFN γ , GzmB) positive cells in CD8⁺ T cells after coculture with the indicated tumor cells. Data are presented as the mean \pm SD.

Data information: n = 3 independent experiments. Data are presented as the mean \pm SD. Statistical significance was determined by one-way ANOVA.

Comments3-4:

PD-L1 expression is also detected not only on tumor cells but also on tumor-infiltrating immune cells such as macrophages and immature myeloid cells. In Fig. 5, was expression levels of PD-L1 expression analyzed only in cancerous but not in noncancerous tissues? The authors should evaluate the expression

levels in each location of histological sections, as in Fig. 1. The results from noncancerous tissues may support the authors claim regarding the intrinsic effect of ERPM1 on PD-L1 expression.

Response: We appreciate the referee for the insightful comments. In Figure 5 in the original manuscript, we performed IHC staining using serial sections of the same HCC tumor tissues to detect the expression of EPDR1, PD-L1 and p65. To address the referee's concern, we performed additional IHC staining and found that EPDR1 expression were increased in HCC lesions compared to adjacent noncancerous tissues (**Figure R9A**). Consistent with this result, western blot analysis showed a markedly increased protein level of EPDR1 in HCC tumor tissues compared with paired noncancerous tissues (**Figure 1B** in the original manuscript, for your convenience, we append the figure here as **Figure R9B**). We further examined the expression levels of EPDR1 and PD-L1 in tumor cells and other cells in the liver from YAP5SA induced HCC-bearing mice. Relative expression analysis by qRT-PCR indicated that PD-L1 was expressed not only in tumor cells but also in immune cells such as macrophages (**Figure R9C**), while EPDR1 was expressed at substantially lower levels in immune cells compared to that in tumor cells (**Figure R9D**). Because of the markedly lower expression level of EPDR1 in adjacent noncancerous tissues, it is difficult to analyze the correlation between the expression of EPDR1 and PD-L1 in noncancerous tissues. Instead, using a series of HCC cell lines and mouse models, we have provided more evidence to verify that EPDR1 increased PD-L1 surface expression in cancer cells (**Figure 3B and C; Figure S3J and E** in the revised manuscript). Collectively, our data indicated that tumor intrinsic EPDR1 promotes tumor immune evasion by enhance PD-L1 expression.

Figure R9. EPDR1 highly expressed in tumor cells and tumor tissues.

- A** Representative immunohistochemistry images of EPDR1 in HCC specimens; scale bars, 100 μ m.
- B** Immunoblotting analysis of EPDR1 protein levels in 20 pairs of clinically matched adjacent noncancerous liver tissues (Normal) and human liver cancer tissues (Tumor). Calnexin served as a loading control.
- C** qRT-PCR analysis of mRNA level PD-L1 in indicated cells in YAP5SA induced mice liver cancer model (n = 3 male, Data are presented as mean \pm SD. Statistical significance was determined by one-way ANOVA).
- D** qRT-PCR analysis of mRNA level EPDR1 in indicated cells in YAP5SA induced mice liver cancer model (n = 3 male. Data are presented as mean \pm SD. Statistical significance was determined by one-way ANOVA).

Comments3-5:

To prove the immunosuppressive effect of EPDR1 on human T cells in clinical setting, association of EPDR1 expression with infiltration of CD8 T cells and/or Treg into tumor sites should be evaluated, as the PD-L1 expression was assessed in Fig. 5.

Response: We thank the referee for the insightful comments. To address this point, we analyzed transcriptomic data from clinical HCC cases in the TCGA/GTEX dataset using the GEPIA website, which revealed that EPDR1 expression was positively correlated with T cell exhaustion, but had no correlation with the infiltration of Treg cells (**Figure R10A**). Further IHC analysis revealed a negative correlation between EPDR1 expression and CD8⁺ T cell population in clinical HCC tissues (**Figure R10B and C**). Nevertheless, multiplex immunohistochemistry analysis of clinical HCC tumor samples showed that the proportion of tumor-infiltrating PD-1⁺CD8⁺ T cells was markedly increased in tumor tissues with higher expression level of EPDR1 compared to that in EPDR1-low tumor tissues (**Figure R10D**). Similar results were observed in YAP5SA-induced HCC mouse model, which revealed that the percentage of tumor-infiltrating PD-1⁺CD8⁺ T cells was significantly higher in mice bearing EPDR1-overexpressing tumor compared to that in control group (**Figure 2D** in the original manuscript). Collectively, these results indicated that EPDR1 in cancer cells impaired the antitumor activity of tumor-infiltrating CD8⁺ T cells. In current study, we did not investigate whether EPDR1 had any effect on the infiltration of Treg cell. Considering the crucial role of Treg cells and the novel function of EPDR1 in regulating tumor immunity, the potential correlation between tumor intrinsic EPDR1 expression and Treg cell function warrants further independent study.

Figure R10. EPDR1 expression is positively correlated with tumor-infiltrating CD8⁺ T cell exhaustion in clinical HCC tumor tissues.

A Analysis of correlation between EPDR1 and different T cell subsets in liver cancer using the TCGA/GTEX dataset.

B Representative immunohistochemistry images of EPDR1 and CD8 staining in HCC specimens. The nucleus was stained with DAPI. scale bars, 50 μm .

C Correlation analysis of EPDR1, CD8⁺ positive signal in HCC specimens, P -values and R were calculated by two-tailed Person's correlation analysis.

D Represent pictures shown CD8⁺PD-1⁺ cells in HCC specimens with different EPDR1 level. The nucleus was stained with DAPI. scale bars as shown.

Comments3-6:

The results of Fig. 1A suggest the patients with higher expression of EPDR exhibit better prognosis. As did in suppl. Fig. S1B, the authors should show the prognosis of EPDR^{high} and EPDR^{low} patients with or without benefit from ICB, to support the authors' claim that higher expression EPDR1 could be a predictive factor for better response to anti-PD-(L)1 therapy.

Response: We truly appreciate the referee for the insightful comments. In fact, this is exactly we expected from this study. Therefore, we screened transcriptomic data from tumor tissues derived from HCC patients with or without responses to ICB, which revealed that EPDR1 was expressed at substantially higher levels in clinical HCC lesion samples from patients with positive response to pembrolizumab (anti-PD-1) therapy compared to that from patients with poor response (**Figure R11A**) (Hong *et al*, 2022). Following our referee's suggestion, we have further analyzed the prognosis of EPDR1^{high} and EPDR1^{low} patients with or without benefit from ICB. The results showed that elevated EPDR1 expression was significantly positively correlated with progression free survival in HCC patients treated with atezolizumab (anti-PD-L1) treatment alone or combined with bevacizumab (anti-VEGF) (**Figure R11B**) (Zhu *et al*, 2022). These data suggested that higher expression EPDR1 might be a predictive factor for better response to anti-PD-(L)1 therapy.

Figure R11. EPDR1 positively correlate with immune checkpoint therapy response among liver cancer patient.

A Analysis of EPDR1 level in responder/non-responder HCC patients of immune checkpoint therapy, Statistical significance was determined t-test, unpaired, one-tailed, source data were obtained from the European Nucleotide Archive (ENA) (primary accession number: PRJEB34724, secondary accession number: ERP117672). Statistical significance was determined by one-tailed unpaired Student's *t*-test.

B Kaplan–Meier analysis of overall survival with log-rank tests for HCC patients receiving immune checkpoint therapy with low versus high expression of EPDR1, source data were obtained from the European Genome-Phenome Archive under accession no. EGAS00001005503.

Comments3-7:

Many types of cancer cells have been reported to express higher levels of EPDR1. Is EPDR1-mediated PD-L1 up-regulation the general immunosuppressive mechanism for cancer progression other than HCC? Or is that specific for HCC? Immunosuppressive effect of EPDR1-overexpressing or knocked down non-HCC cancer lines *in vivo* (Nude/ B6) can be evaluated in syngeneic rather than xenograft Hepa 1-6 model. This issue is critical for considering broad application for many types of cancer.

Response: We appreciate the referee's suggestion. To address this point, we analyzed the correlation between EPDR1 and PD-L1 expression in various clinical tumor tissues in the TCGA/GTEX dataset. The results showed that EPDR1 expression levels were positively correlated with PD-L1 transcriptional levels in pheochromocytoma and paraganglioma (PCPG), kidney renal papillary cell carcinoma (KIRP), and liver hepatocellular carcinoma (LICH), while the correlation between EPDR1 and PD-L1 expression was weak in lung adenocarcinoma (LUAD) and colon adenocarcinoma (CHOL) (**Figure R12A**). Considering that, we would like to request not to do this *in vivo* experiment since it's impossible for us to get these murine cell lines and accomplish the *in vivo* experiments during the time frame allowed for this revision. Instead, we examined PD-L1 expression in human cancer cell lines with EPDR1 knockdown or overexpression. Western blotting analysis confirmed that EPDR1 knockdown decreased, while overexpression increased the protein levels of PD-L1 in RCC4 and PANC1 cells (**Figure R12B**). Consistent with TCGA data, either EPDR1 knockdown or overexpression had no effect on PD-L1 expression in PC9 and HCT116 cells (**Figure R12C**). These data suggested that EPDR1-mediated PD-L1 upregulation was not the general immunosuppressive mechanism for all cancer progression. We guess there might be multiple reasons responsible for this, such as the differential expression levels and activity of EPDR1, TRIM21 and NF- κ B. Nevertheless, following all the insightful comments from our referees, we have now improved the study substantially and strength the evidence for that EPDR1 promotes PD-L1 expression and tumor immune evasion by inhibiting TRIM21-dependent ubiquitylation of IKBKB.

Figure R12. EPDR1 promotes PD-L1 expression in RCC4 and PANC1 cells.

- A** Scatter diagram analysis of the expression correlation of EPDR1 and PD-L1 in multiple cancers using the TCGA/GTEX dataset.
- B, C** Western blot analysis of EPDR1 and PD-L1 levels in multiple cell lines, include pancreatic cancer (PANC1), kidney cancer (RCC4) in (B), lung cancer (PC9) and colon cancer (HCT116) in (C), β -actin was used as a control.

Comments3-8:

Any data demonstrating that ERPM1-mediated regulation of TRIM21 activity really contribute to the up-regulation of PD-L1 on tumor cells *in vivo* are missing. The comparative analysis of PD-L1 expression on the cell surface should be shown in both ERPM1-overexpressing (+/- TRIM21 co-transfection) and ERPM1-knocked down tumor cells with flow cytometric analysis.

Response: We appreciated the referee for the insightful comments. Following the referee's suggestion, we performed additional *in vivo* experiments to demonstrate that EPDR1-mediated regulation of TRIM21 activity contributes to the upregulation of PD-L1 on tumor cells. By employing a xenograft model mice generated by subcutaneous inoculation with Hepa 1-6 cells with differential expression levels of EPDR1 and TRIM21, we observed that EPDR1 overexpression facilitated the *in vivo* tumor growth of Hepa 1-6 xenografts, which was largely abolished by TRIM21 overexpression (**Figure R5**, Referee #2 also raised a correlated question. For your convenience, we append the figures here once again). We also found that EPDR1 overexpression increased, while TRIM21 overexpression decreased, the surface expression of PD-L1 on tumor cells (**Figure R5D**). Further flow-cytometric analysis revealed that TRIM21 expression could reverse the increased exhaustion and decreased activity of tumor-infiltrating T cells in mice inoculated with Hepa 1-6 cells overexpressing EPDR1 (**Figure R5E and F**). Collectively, these results indicated that EPDR1-mediated regulation of TRIM21 activity was necessary for upregulation of surface PD-L1 expression and tumor-infiltrating CD8⁺ T cell exhaustion.

Figure R5. TIM21 is involved in EPDR1-mediated tumor immune evasion.

A-C Hepa 1-6 cells stably expressing Flag-EV, Flag-mTRIM21, Flag-mEPDR1 or Flag-mTRIM21 plus Flag-mEPDR1 both were injected subcutaneously into C57BL/6J mice (n = 6 male mice per group). Tumor size was measured starting at 10 days after inoculation. Figures depict xenografts (A), growth curves (B) and tumor weight (C) determined at the end of the experiment (day 25). Data are presented as the mean \pm SEM.

D Flow cytometric analysis of membrane-bound PD-L1 in the indicated group in (A). Data are presented as the mean \pm SD.

E Flow cytometric analysis of ratio of immunosuppressive molecules (PD-1, TIM-3) positive cells in tumor CD8⁺ T cells from the indicated group in (A). Data are presented as the mean \pm SD.

F Flow cytometric analysis of ratio of immune effector molecules (IFN γ , GzmB) positive cells in tumor CD8⁺ T cells from the indicated group in (A). Data are presented as the mean \pm SD.

Data information: Statistical significance was determined by two-way ANOVA (B) and one-way ANOVA (C-F).

We also analyzed the surface expression of PD-L1 on tumor cells in xenograft model mice generated by subcutaneous inoculation with EPDR1 knockdown or NTC Hepa 1-6 cells. Tumor growth was suppressed in mice inoculated with EPDR1 knockdown Hepa 1-6 cells (**Fig R13 A-C**). Importantly, the surface expression of PD-L1 was reduced on EPDR1 knockdown tumor cells compared to that on NTC

tumor cells (**Figure R13 D**). Furthermore, the infiltrated CD8⁺ T cells exhibited lower PD-1 and TIM-3 expression and higher IFN γ and GzmB expression in mice inoculated with EPDR1 knockdown Hepa 1-6 cells compared with those in NTC-inoculated mice (**Figure R13E and F**). These data indicated that loss of EPDR1 in cancer cells resulted in reduced PD-L1 surface expression and enhanced the antitumor activity of tumor-infiltrating CD8⁺ T cells.

Figure R13. EPDR1 knockout suppresses HCC tumor growth and tumor-infiltrating CD8⁺ T cell exhaustion.

A-C Hepa 1-6 cells stably expressing NTC or shEPDR1 were injected subcutaneously into C57BL/6J mice (n = 6 male mice per group). Tumor size was measured starting at 10 days after inoculation. Figure depict xenografts (A), growth curves (B) and tumor weights (C) determined at the end of the experiment (day 25). Data are presented as the mean \pm SEM.

D Flow cytometric analysis of membrane-bound PD-L1 in the indicated group in (A). Data are presented as the mean \pm SD.

E Flow cytometric analysis of ratio of immunosuppressive molecules (PD-1, TIM-3) positive cells in tumor CD8⁺ T cells from the indicated group in (A). Data are presented as the mean \pm SD.

F Flow cytometric analysis of ratio of immune effector molecules (IFN γ , Granzyme B) positive cells in tumor CD8⁺ T cells from the indicated groups in (A). Data are presented as the mean \pm SD.

Data information: n = 6 male mice per group, mean \pm SD, Statistical significance was determined by two-way ANOVA (B) and one-way ANOVA (D-F).

Comments3-9:

Regulation of TRIM21 level alone modulated IKBKB ubiquitination in vitro (Fig. 3 and FigS3) and seemed to ameliorate the exhausted phenotype of CD8 T cells in vivo (Fig. 4). Did overexpression of TRIM21 alone affect the PD-L1 expression on the surface, and other immunosuppressive molecules? This is related to the comment #1.

Response: Following the referee's suggestion, we investigate the expression of PD-L1 on the surface and other immunosuppressive molecules in HepG2 cells overexpressing TRIM21. Flow-cytometric analysis showed that the cell surface expression of membrane bound PD-L1 was decreased in HepG2 cells overexpressing TRIM21 compared to that in EV group (**Figure R14A**). Relative expression analysis by qRT-PCR indicated that TRIM21 overexpression significantly reduced the expression levels of PD-L1, IL-6 IL-8 and IL-1 β . Consistent with these results (**Figure R14B**), EPDR1 overexpression significantly increased the surface expression of PD-L1 and transcription of PD-L1, IL-6 and IL-1 β (**Figure R7**). Moreover, our Co-IP assays and Pull-down assays confirmed the interaction between EPDR1 and TRIM21 (**Figure 3J and K** in the revised manuscript). More importantly, TRIM21 expression could reverse the increased exhaustion and decreased activity of tumor-infiltrating T cells in mice inoculated with Hepa 1-6 cells overexpressing EPDR1 (**Figure 3J and K** in the revised manuscript). Collectively, these data suggested that interaction with TRIM21 was necessary for EPDR1-mediated PD-L1 transactivation and tumor-infiltrating CD8⁺ T cell exhaustion.

Figure R14. TRIM21 overexpression suppresses the expression of surface PD-L1 and NF- κ B downstream factors.

A Flow cytometric analysis of PD-L1 cell surface expression in HepG2 cells expressing Flag-TRIM21 or Flag-EV.

B qRT-PCR analysis of mRNA level of NF- κ B pathway target genes in HepG2 cells with the indicated genotypes.

Data information: n = 3 independent experiments and data are presented as the mean \pm SD. Statistical significance was determined by two-tailed unpaired Student's *t*-test (A) and two-way ANOVA (B).

Reference

Gullà A, Hideshima T, Bianchi G, Fulciniti M, Kemal Samur M, Qi J, Tai YT, Harada T, Morelli E, Amodio N, Carrasco R, Tagliaferri P, Munshi NC, Tassone P, Anderson KC (2018) Protein arginine methyltransferase 5 has prognostic relevance and is a druggable target in multiple myeloma. *Leukemia* 32: 996-1002

- Han Y, Zhang Y-Y, Pan Y-Q, Zheng X-J, Liao K, Mo H-Y, Sheng H, Wu Q-N, Liu Z-X, Zeng Z-L, Yang W, Yuan S-Q, Huang P, Ju H-Q, Xu R-H (2023) IL-1 β -associated NNT acetylation orchestrates iron-sulfur cluster maintenance and cancer immunotherapy resistance. *Molecular Cell* 83: 1887-1902.e1888
- Hong JY, Cho HJ, Sa JK, Liu X, Ha SY, Lee T, Kim H, Kang W, Sinn DH, Gwak GY, Choi MS, Lee JH, Koh KC, Paik SW, Park HC, Kang TW, Rhim H, Lee SJ, Cristescu R, Lee J et al. (2022) Hepatocellular carcinoma patients with high circulating cytotoxic T cells and intra-tumoral immune signature benefit from pembrolizumab: results from a single-arm phase 2 trial. *Genome Med* 14: 1
- Huang Y, Gao X, He QY, Liu W (2024) A Interacting Model: How TRIM21 Orchestrates with Proteins in Intracellular Immunity. *Small Methods* 8: e2301142
- Johnson DE, O'Keefe RA, Grandis JR (2018) Targeting the IL-6/JAK/STAT3 signalling axis in cancer. *Nature Reviews Clinical Oncology* 15: 234-248
- Kersten K, Hu KH, Combes AJ, Samad B, Harwin T, Ray A, Rao AA, Cai E, Marchuk K, Artchoker J, Courau T, Shi Q, Belk J, Satpathy AT, Krummel MF (2022) Spatiotemporal co-dependency between macrophages and exhausted CD8+ T cells in cancer. *Cancer Cell* 40: 624-638.e629
- Ma X, Bi E, Lu Y, Su P, Huang C, Liu L, Wang Q, Yang M, Kalady MF, Qian J, Zhang A, Gupte AA, Hamilton DJ, Zheng C, Yi Q (2019) Cholesterol Induces CD8+ T Cell Exhaustion in the Tumor Microenvironment. *Cell Metabolism* 30: 143-156.e145
- Ribas A, Wolchok JD (2018) Cancer immunotherapy using checkpoint blockade. *Science* 359: 1350-1355
- Tang T, Huang X, Lu M, Zhang G, Han X, Liang T (2023) Transcriptional control of pancreatic cancer immunosuppression by metabolic enzyme CD73 in a tumor-autonomous and -autocrine manner. *Nat Commun* 14: 3364
- Wada K, Niida M, Tanaka M, Kamitani T (2009) Ro52-mediated Monoubiquitination of IKK Down-regulates NF- κ B Signalling. *Journal of Biochemistry* 146: 821-832
- Zhang Y, Wang M, Ye L, Shen S, Zhang Y, Qian X, Zhang T, Yuan M, Ye Z, Cai J, Meng X, Qiu S, Liu S, Liu R, Jia W, Yang X, Zhang H, Zhong X, Gao P (2024) HKDC1 promotes tumor immune evasion in hepatocellular carcinoma by coupling cytoskeleton to STAT1 activation and PD-L1 expression. *Nat Commun* 15: 1314
- Zhu AX, Abbas AR, de Galarreta MR, Guan Y, Lu S, Koeppen H, Zhang W, Hsu CH, He AR, Ryoo BY, Yau T, Kaseb AO, Burgoyne AM, Dayyani F, Spahn J, Verret W, Finn RS, Toh HC, Lujambio A, Wang Y (2022) Molecular correlates of clinical response and resistance to atezolizumab in combination with bevacizumab in advanced hepatocellular carcinoma. *Nat Med* 28: 1599-1611

Dear Dr Ping Gao,

Thank you for submitting your revised manuscript (EMBOJ-2023-116324R) to The EMBO Journal. Your amended study was sent back to the three referees for their scientific re-evaluation, and we have received detailed comments from all of them, which I enclose below. As you will see, the experts state that the work has been substantially improved by the revisions and they are now broadly in favour of publication, pending minor amendments.

Thus, we are pleased to inform you that your manuscript has been accepted in principle for publication in The EMBO Journal.

Please consider the remaining minor concerns of referee #1 carefully and adjust the data presentation and discussion of the findings where appropriate.

We also now need you to take care of a number of issues related to formatting and data annotation as detailed below, which should be addressed at re-submission.

Please contact me at any time if you have additional questions related to below points.

As you might remember from our previous exchange, every paper at the EMBO Journal now includes a 'Synopsis', displayed on the html and freely accessible to all readers. The synopsis includes a 'model' figure as well as 2-5 one-short-sentence bullet points that summarize the article. I would appreciate if you could provide this figure and the bullet points.

Thank you for giving us the chance to consider your manuscript for The EMBO Journal. I look forward to your final revision.

Again, please contact me at any time if you need any help or have further questions.

Best regards,

Daniel Klimmeck

>> Authors: Please define all corresponding authors on the manuscript's first page.

>> Author Contributions: Please remove the author contributions information from the manuscript text. Note that CRediT has replaced the traditional author contributions section as of now because it offers a systematic machine-readable author contributions format that allows for more effective research assessment. and use the free text boxes beneath each contributing author's name to add specific details on the author's contribution.

More information is available in our guide to authors.

>> Figures: the main figures should be uploaded as individual, high resolution figure files. The figure legends should be removed from the main figures and added to the manuscript text;

>> Figure callouts: Figure 3H needs to be called out.

>> References: adjust reference format to EMBO Journal format, 10 authors et al, and place References after the Discussion, before figure legends.

>> Please provide source data for the study as to the separate request e-mail by my colleague Hannah Sonntag.

>> Section order should be corrected as follows: title page with complete author information, abstract, keywords, introduction, results, discussion, methods, data availability section, acknowledgements, disclosure and competing interests statement, references, main figure legends, tables, expanded figure legends.

>> Reference your 2023 article (PMID: 36934105) in the Methods section.

>> Data availability section: remove referee token for the GEO dataset and make sure privacy is released.

>> Appendix formatting: The file should be renamed "Appendix" and the figures should be renamed "Appendix Figure S1, S2" etc. . The tables be removed and combined into a reagents and tools table - please compare the template in our authors guidelines, and it should be uploaded separately. The table of contents of the appendix needs pages numbers added.

>>Dataset EV legends: Table S1 should be renamed "Dataset EV1" and a legend should be added to the file in a separate worksheet.

>> Please recheck data redisplay of FACS images in the figures EV5E, IFNg vs GranzmB, EV.

>> Consider additional changes and comments from our production team as indicated below:

- DAS:

1. Please note that the specific URL for GSE250169 dataset is not provided in the data availability statement.
2. Please note that reviewer access code for GSE250169 dataset is not provided in the data availability statement.
3. Please note that the exact p values are not provided in the legends of figures 2e; 3a; 4b, g, i-k; 5b-c; EV 1a; EV 2g-h; EV 3d; EV 4b; EV 5b, d-e; EV 6a.
4. Please indicate the statistical test used for data analysis in the legends of figures 2b; 3d; EV 4g.
5. Please note that the box plots need to be defined in terms of minima, maxima, centre, bounds of box and whiskers, and percentile in the legends of figures 1c, e; EV 1a.
6. Please note that information related to n is missing in the legends of figures 1c, e; 2d; 3b-c; EV 2d, i-j; EV 4h-i.
7. Please note that the error bars are not defined in the legends of figures 3b-c; EV 3d.
8. Please note that the "+" is not defined in the legend of figure 3k, m; EV 3l. This needs to be rectified.

Referee #1:

The manuscript investigates the role of EPDR1 in PD-L1 regulation. The authors show, EPDR1 interacts with TRIM21 that promotes an IKBKB-NF-κB dependent transcription of PD-L1. In the revised manuscript, the authors have performed extensive work to answer the questions raised by the reviewers. The work is considered satisfactory for manuscript acceptance. Minor concerns include analysis of PD1+TIM3+ double positive CD8 T cell population be added to the figures. Figure R2 should be added in the supplement and the authors should consider revising its title since shEPDR1 group is not significantly different than anti-PDL1 alone.

Referee #2:

They have addressed my previous comment, and I have no more new comments.

Referee #3:

In this revised manuscript, several experimental evaluations have been performed by the authors based on the reviewers' criticisms. The authors have addressed my concerns, and the revised manuscript has been improved.

Point-by-point response

Referee #1:

The manuscript investigates the role of EPDR1 in PD-L1 regulation. The authors show, EPDR1 interacts with TRIM21 that promotes an IKBKB-NF- κ B dependent transcription of PD-L1. In the revised manuscript, the authors have performed extensive work to answer the questions raised by the reviewers. The work is considered satisfactory for manuscript acceptance. Minor concerns include analysis of PD1⁺TIM3⁺ double positive CD8 T cell population be added to the figures. Figure R2 should be added in the supplement and the authors should consider revising its title since shEPDR1 group is not significantly different than anti-PDL1 alone.

Response: We are grateful for the referee's suggestions. To address the concerns and maintain the original data layout, we supply additional analysis of PD1⁺TIM3⁺ CD8 T cell population of YAP5SA-induced HCC model, as shown in FigEV2C. The results are consistent with our conclusion and indicate that EPDR1 facilitates exhaustion of CD8⁺ T cells. In additional, following the suggestion of referee#1, we have added the Figure R2 to the Appendix Figure S2 and have described it in the results.

Referee #2:

They have addressed my previous comment, and I have no more new comments.

Response: Thank you!

Referee #3:

In this revised manuscript, several experimental evaluations have been performed by the authors based on the reviewers' criticisms. The authors have addressed my concerns, and the revised manuscript has been improved.

Response: Thank you!

Dear Dr Ping Gao,

Thank you for submitting the revised version of your manuscript. I have now evaluated your amended manuscript and concluded that the remaining minor concerns have been sufficiently addressed.

I am thus pleased to inform you that your manuscript has been accepted for publication in the EMBO Journal.

Also, we kindly ask for your consent on keeping the referee figures included in this file.

On a different note, I would like to alert you that EMBO Press offers a format for a video-synopsis of work published with us, which essentially is a short, author-generated film explaining the core findings in hand drawings, and, as we believe, can be very useful to increase visibility of the work. Please see the following link for representative examples and their integration into the article web page:

<https://www.embopress.org/doi/full/10.15252/emj.2019103932>

If you have any questions, please do not hesitate to contact the Editorial Office.

Best regards,

Daniel Klimmeck

Daniel Klimmeck, PhD
Senior Editor
The EMBO Journal
EMBO
Postfach 1022-40
Meyerhofstrasse 1
D-69117 Heidelberg
contact@embojournal.org